# Cell type boundaries organize plant development

**Monica Pia Caggiano[1†], Xiulian Yu[1†], Neha Bhatia[1,2†], André Larsson[3], Hasthi Ram[1], Carolyn K Ohno[1,2], Pia Sappl[1], Elliot M Meyerowitz[4], Henrik Jönsson[3,5,6], Marcus G Heisler[1,2]\***

[1]European Molecular Biology Laboratory, Heidelberg, Germany; [2]School of Life and Environmental Sciences, University of Sydney, Sydney, Australia; [3]Computational Biology and Biological Physics, Department of Astronomy and Theoretical Physics, Lund University, Lund, Sweden; [4]Division of Biology and Biological Engineering, California Institute of Technology, Howard Hughes Medical Institute, Pasadena, United States; [5]Sainsbury Laboratory, University of Cambridge, Cambridge, United Kingdom; [6]Department of Applied Mathematics and Theoretical Physics, University of Cambridge, Cambridge, United Kingdom

**\*For correspondence:**
marcus.heisler@sydney.edu.au

[†]These authors contributed equally to this work

**Competing interests:** The authors declare that no competing interests exist.

**Abstract** In plants the dorsoventral boundary of leaves defines an axis of symmetry through the centre of the organ separating the top (dorsal) and bottom (ventral) tissues. Although the positioning of this boundary is critical for leaf morphogenesis, how the boundary is established and how it influences development remains unclear. Using live-imaging and perturbation experiments we show that leaf orientation, morphology and position are pre-patterned by HD-ZIPIII and KAN gene expression in the shoot, leading to a model in which dorsoventral genes coordinate to regulate plant development by localizing auxin response between their expression domains. However we also find that auxin levels feedback on dorsoventral patterning by spatially organizing HD-ZIPIII and KAN expression in the shoot periphery. By demonstrating that the regulation of these genes by auxin also governs their response to wounds, our results also provide a parsimonious explanation for the influence of wounds on leaf dorsoventrality.
DOI: https://doi.org/10.7554/eLife.27421.001

## Introduction

Lateral organ development in plants and animals typically involves several processes occurring in a coordinated manner. These include organ positioning, the specification of different cell types and organ morphogenesis. Spatial cues specifying these processes are usually provided by a molecular pre-pattern present in precursor tissues, or from inductive signals emanating from neighboring regions. Unlike animals however, plant organs such as leaves arise continuously in regular patterns around the shoot apical meristem (SAM). Nevertheless, certain features of leaves are relatively constant including the restriction of their formation to the meristem periphery and their flattened, dorsoventral (top-bottom) orientation with respect to the shoot apex. How are these fundamental features specified?

Since the 1950s wounding experiments involving the isolation of leaf primordia from the meristem have suggested the presence of an inductive signal from the meristem that promotes dorsal identity within leaf primordia at the time of organ initiation (*Reinhardt et al., 2005*; *Sussex, 1951*). A variant on this theme is the proposal that transient auxin depletion in the adaxial (adjacent to the shoot axis) tissues of leaf primordia promotes dorsal identity (*Qi et al., 2014*). In contrast, other studies suggest that dorsoventral patterning is pre-established, being directly derived from central-peripheral patterning of the shoot (*Hagemann and Gleissberg, 1996*; *Husbands et al., 2009*;

*Kerstetter et al., 2001*; *Koch and Meinhardt, 1994*). Supporting this proposal is the observation that transcription factors involved in both dorsal and ventral cell fate in the leaf are expressed in the SAM in a central and peripheral manner respectively (*Emery et al., 2003*; *Kerstetter et al., 2001*; *McConnell et al., 2001*; *Yadav et al., 2013*). Thus, the manner in which leaf dorsoventrality is first established in leaves remains unresolved (*Kuhlemeier and Timmermans, 2016*).

To explain how leaf dorsoventrality regulates morphogenesis, *Waites and Hudson, 1995* took a cue from wing development in Drosophila (*Diaz-Benjumea and Cohen, 1993*) and proposed that tissues located along the boundary between dorsal and ventral leaf tissues, i.e. along the leaf margin, act as organizers by producing mobile signals that pattern lamina growth (*Waites and Hudson, 1995*). So far, several genes have been identified that are expressed along this boundary including KLUH (*Anastasiou et al., 2007*) and the WOX family transcription WOX1, WOX2 and PRS (*Haecker et al., 2004*; *Matsumoto and Okada, 2001*; *Nakata et al., 2012*). The WOX genes in particular are required for proper lamina growth but are expressed only at the boundary region suggesting they may promote long-range patterning (*Nakata et al., 2012*). However although their ectopic expression can cause filamentous outgrowths at the leaf base, overall lamina growth is fairly normal (*Nakata et al., 2012*) suggesting that additional factors must direct patterning. The WOX genes are also known to regulate the expression of both dorsal and ventrally expressed transcription factors and miRNAs (*Nakata et al., 2012*; *Zhang et al., 2014*; *Zhang et al., 2017*) indicating that the WOX genes promote lamina growth at least in part by maintaining the integrity of dorsal and ventral expression domains. Overall, this leaves the question of how dorsoventral boundaries actually regulate morphogenesis still unanswered.

Besides influencing leaf differentiation and shape, genes involved in leaf dorsoventrality influence leaf position. For instance, Arabidopsis plants mutant for the KANADI (KAN) genes develop leaves ectopically from the hypocotyl and leaf tissues (*Izhaki and Bowman, 2007*) while plants mutant for the Class III HD-ZIP (HD-ZIPIII) genes develop leaves from the center of the shoot (*Emery et al., 2003*). These observations indicate that the developmental mechanisms that specify leaf dorsoventrality may also be involved in organ positioning although how these processes relate is unclear.

In this study we investigate the origin of dorsoventral patterning in detail. We show that new organs are centered on a pre-patterned boundary region located between the expression domains of HD-ZIPIII and KAN genes and that both KAN and HD-ZIPIII genes act to repress organogenesis where they are expressed. This leads to a model in which dorsoventral genes control leaf morphogenesis and positioning by localizing auxin transcriptional response, which then polarizes cells non-cell autonomously (*Bhatia et al., 2016*). However, we also show that dynamic auxin levels play a central role in determining boundary position by spatially organizing HD-ZIPIII and KAN1 expression in the meristem periphery. By demonstrating that the regulation of these genes by auxin also governs their response to wounds, our results also provide a parsimonious explanation for the influence of wounds on leaf dorsoventrality.

## Results

### Genes involved in leaf dorsoventrality pre-pattern organ formation in the shoot

To better understand how leaf dorsoventrality is first established we used confocal microscopy to monitor the expression of genes involved in leaf dorsoventrality in the shoot apical meristem (SAM) in combination with the auxin efflux carrier PIN-FORMED1 (PIN1) using functional fluorescent protein reporters. In general, we found proteins involved in leaf dorsoventrality to be localized in non-overlapping concentric patterns with the dorsal Class III HD-ZIP protein REVOLUTA (REV) (*Otsuga et al., 2001*) being detected centrally, as marked by the expression of REVp::REV-2 × YPet (REV-2 × YPet), and the ventral protein KANADI1 (KAN1) (*Kerstetter et al., 2001*) expressed peripherally, as marked by the expression of KAN1p::KAN1−2 × GFP (KAN1−2 × GFP) (*Figure 1*, A to E). These concentric domains not only encircled the meristem but also extended contiguously around initiating leaves and floral bracts. Imaging PIN1p::PIN1-CFP (PIN1-CFP) (*Gordon et al., 2007a*) together with these reporters revealed regions of high PIN1-CFP expression, marking positions of organ inception (*Heisler et al., 2005*; *Reinhardt et al., 2003*), to be centered on a narrow region or 'gap' located between the dorsal and ventral domains (*Figure 1D–K*). As found previously, REV expression

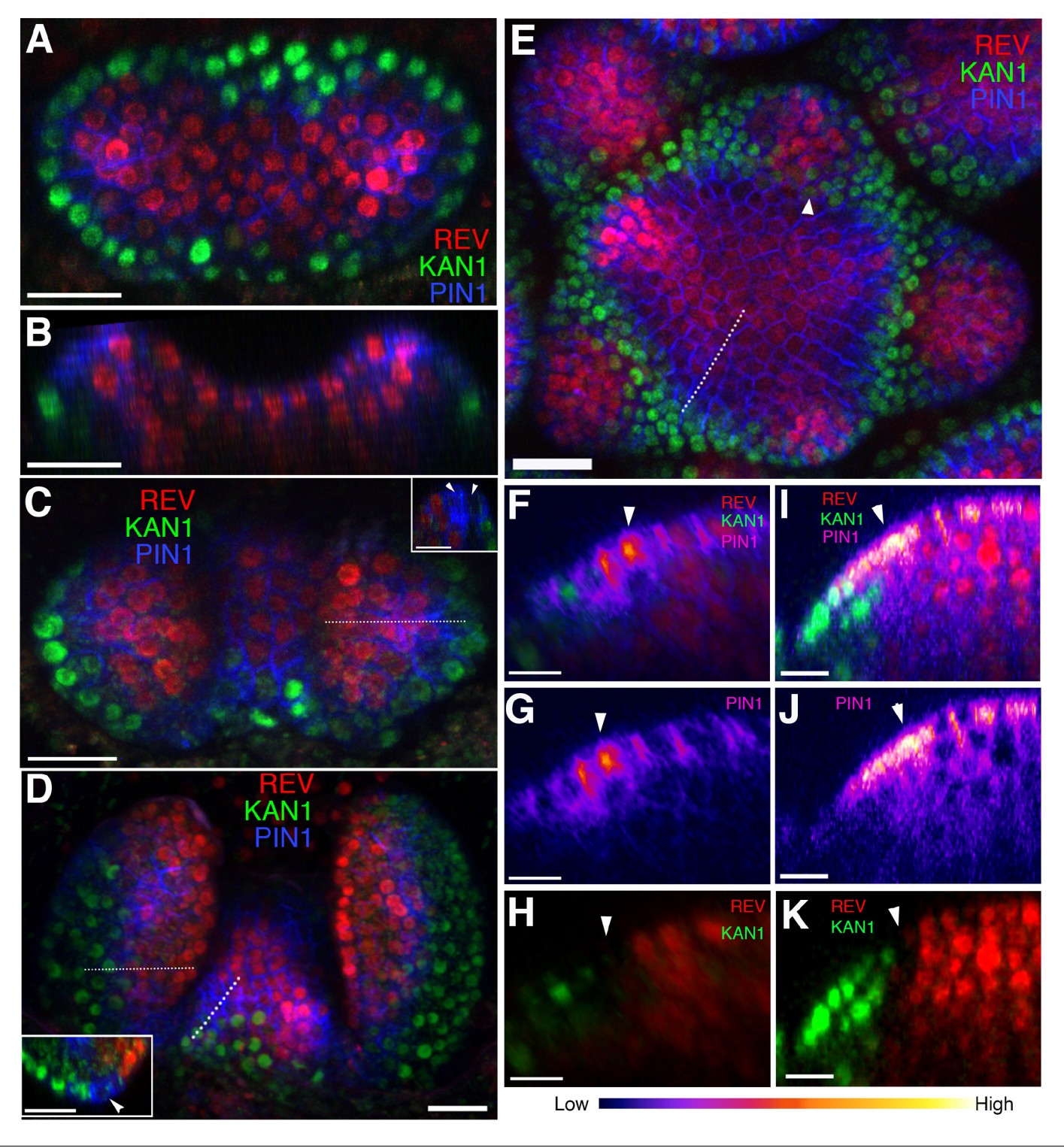

**Figure 1.** Organ initiation is centered on a boundary between the expression domains of genes involved in leaf dorsoventrality. (A–D) Confocal projections showing REV-2 ×YPet (red), PIN1-CFP (blue) and KAN1−2 × GFP (green) expression in a vegetative shoot apical meristem at 3 days (A), 4 days (C) and 5 days (D) after stratification (DAS), respectively. Inset in (C) shows a longitudinal optical section across the dotted line in the right leaf. Inset in (D) shows a transverse optical section across the dotted line in the left leaf. White arrowheads in the insets in (C and D) mark a gap in between the REV-2 ×YPet and KAN1−2 × GFP expression domains where PIN1 expression is highest. (B) Longitudinal reconstructed section of seedling shown in (A). (E) Expression pattern of REV-2 ×YPet, KAN1−2 × GFP and PIN1-CFP in an inflorescence meristem. White arrow head marks region where

*Figure 1 continued on next page*

*Figure 1 continued*

KAN1−2 × GFP expression is being reestablished after floral bract emergence. (**F–K**) Longitudinal optical sections across the dashed lines in (**D**) and (**E**) showing localized PIN1-CFP expression (magenta) marking organ inception at the REV-2 ×YPet/KAN1−2 × GFP boundary in both the vegetative meristem (**F–H**) and inflorescence meristem (**I–K**). White arrow heads mark cells in between the REV-2 ×YPet and KAN1−2 × GFP expression domains where PIN1-CFP expression is highest. Scale bars, 20 μm (A-E, inset in (**D**)) and 10 μm (**F–K**).

DOI: https://doi.org/10.7554/eLife.27421.002

The following figure supplement is available for figure 1:

**Figure supplement 1.** REV and KAN1 display dynamic expression pattern during different stages of organ development.

DOI: https://doi.org/10.7554/eLife.27421.003

extended out into the PIN1-marked primordia over time (*Heisler et al., 2005*), reducing the 'gap' between the REV and KAN domains (*Figure 1—figure supplement 1A–F,M–P*). Nevertheless, at later stages a more substantial 'gap' reappeared between the REV and KAN1 domains (*Figure 1C and D* insets; *Figure 1—figure supplement 1G–L and Q–R*) indicating that the distance between the REV and KAN1 domains can change dynamically. Time-lapse imaging revealed that few cells alter their KAN1 expression during these early stages (*Figure 2A–F*) suggesting that the boundary of KAN1 expression in the SAM marks cells destined to form the future dorsoventral boundary of the organ.

Ventrally expressed MIR165/166 (*Kidner and Martienssen, 2004*; *Merelo et al., 2016*; *Nogueira et al., 2007*; *Yao et al., 2009*) also appeared active in the SAM periphery as marked by a *MIR166Ap::GFPER* reporter and MIR165/166 biosensor (*Figure 3A–I*), consistent with previous studies (*Miyashima et al., 2013*). Both KAN1−2 × GFP and *MIR166Ap::GFPER* re-established their expression in the SAM periphery after organ outgrowth (*Figure 1E* arrowhead; *Figure 1—figure supplement 1I,K and L* and *Figure 3C* arrowhead). As members of the WOX family of transcription factors including WOX1 and PRS are expressed at dorsoventral boundaries (*Nakata et al., 2012*) we imaged functional translational fusions to both PRS (PRSp::PRS- 2 × GFP) and WOX1 (WOX1p:: 2 × GFP-WOX1) to examine their expression in the shoot relative to leaves. We found that although 2 × GFP-WOX1 expression was limited to the margins and middle domain of leaves (*Figure 4A and B*) and absent in the inflorescence meristem (*Figure 4C*), in the vegetative shoot PRS-2 ×GFP expression extended from the leaf middle domain to surround the SAM in a region between the KAN1 and REV expression domains (*Figure 4D–H*). In the inflorescence meristem however, PRS-2 × GFP expression was restricted to early cryptic bract primordia (*Figure 4I and J*). In contrast to all genes described so far, a FILp::dsRED-N7 marker was expressed in both the abaxial (away from the shoot axis) and adaxial regions of the primordium at inception, consistent with a previous study (*Tameshige et al., 2013*) (*Figure 5*). Overall these data reveal that in many respects although not all, dorsoventral patterning within young leaf primordia, including the middle region (*Nakata et al., 2012*), corresponds to and is contiguous with, the patterning of dorsoventral gene expression in the SAM.

## The KAN and class III HD-ZIP genes repress organ initiation where they are expressed

The finding that sites of organ inception marked by PIN1 are located between KAN1 and REV expression domains in the SAM suggest that both KAN1 and REV may act to repress organ inception where they are expressed. Supporting this proposal, leaves develop ectopically in *kan1 kan2 kan4* and *rev phb phv* mutants (*Izhaki and Bowman, 2007*). However it is possible that the HD-ZIPIII genes influence organ initiation only indirectly by promoting SAM formation during embryogenesis (*Emery et al., 2003*; *Izhaki and Bowman, 2007*). To distinguish between these possibilities we induced the expression of MIR165/166-resistant REVr-2 ×VENUS throughout the epidermis using a pOp6/LhGR two-component (*Samalova et al., 2005*) system with the ATML1 promoter (*Sessions et al., 1999*) driving LhGR and found that it caused an arrest of organ formation and repression of KAN1−2 × GFP in both the vegetative and inflorescence meristems (*Figure 6A–E*). Despite the lack of organs, stem growth for the inflorescence meristem continued without any obvious change to meristem size, indicating that the phenotype initially influenced organ initiation (inset in *Figure 6E*). Similar results were observed after induction of a short tandem target mimicry construct designed to repress MIR165/166 activity (*Yan et al., 2012*) (*Figure 6F and G*) or after

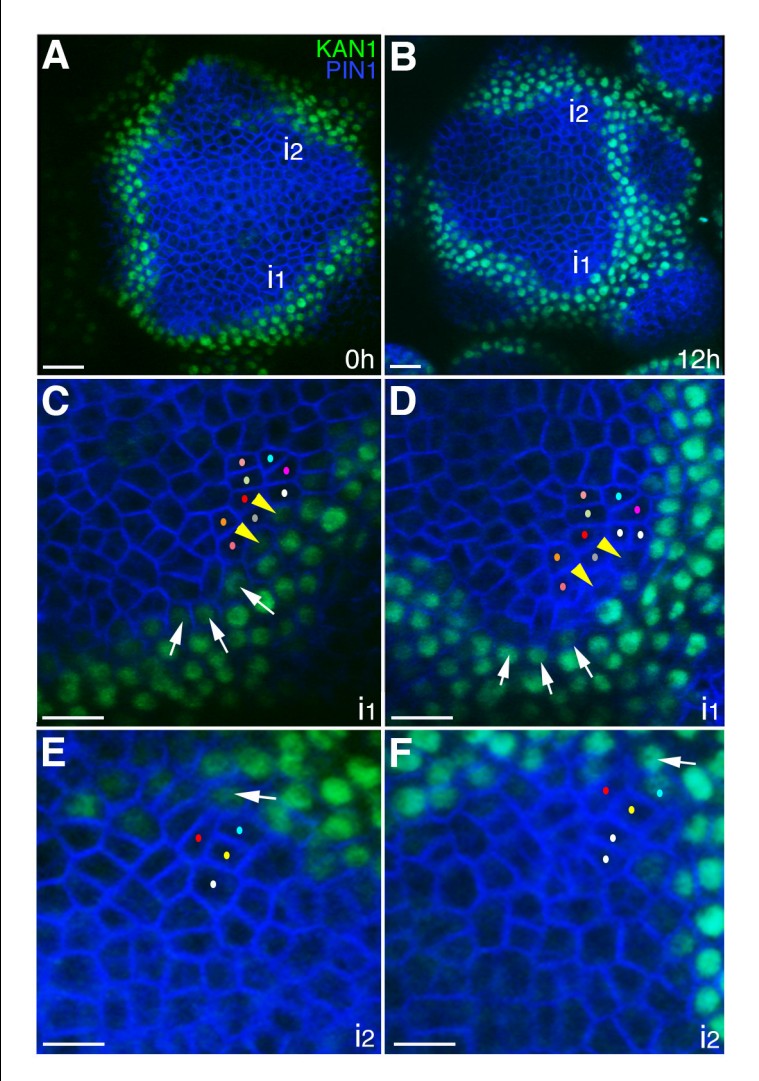

**Figure 2.** The expression of KAN1−2 × GFP is relatively stable with respect to the underlying cells within initiating organs. (A–B) Confocal projections showing an inflorescence meristem viewed from above expressing PIN1-CFP (blue) and KAN1−2 × GFP (green) at two time points (0 hr and 12 hr). Two incipient primordia are marked i1 and i2. (C–D) Close up views corresponding to primordium i1 from (A) and (B) with white arrows marking three cells at the edge of KAN1−2 × GFP expression that retain this expression over the time interval. Yellow arrow heads mark two cells in which KAN1−2 × GFP is absent at 12 hr. Similar colored dots mark the same cells in (C) and (D), tracked over 12 hr. (E) Close up view of primordium I2 in (A) with arrow marking adaxial edge of KAN1 expression. (F) Close up of primordium I2 in (B) showing the same cell marked in (E) remaining at the adaxial edge of KAN1 expression after 12 hr. Similar colored dots mark the same cells in (E) and (F), tracked over 12 hr. Scale bars, 20 µm in (A and B); 10 µm in (C–F).
DOI: https://doi.org/10.7554/eLife.27421.004

epidermal induction of MIR165/166-resistant PHAVOLUTA (*Figure 6H–J*). Similarly, plants express-ing KAN1 ectopically in the epidermis also stopped making new organs (*Figure 6K and L*) and the inflorescence meristem took on a dome shape before eventually arresting (*Figure 6M*). We conclude both the KAN and Class III HD-ZIP genes regulate organ positioning at least in part by repressing organ initiation where they are expressed.

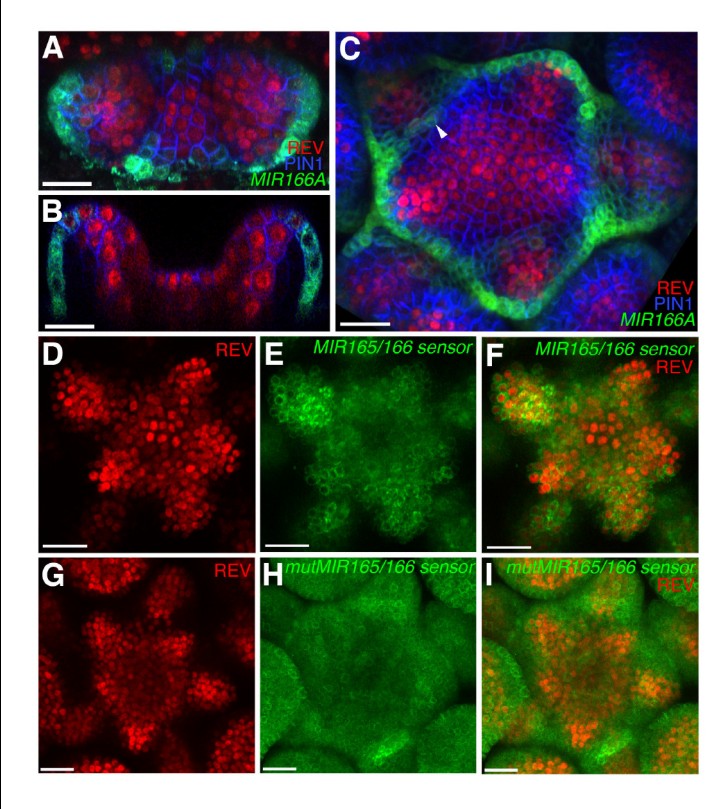

**Figure 3.** Expression and activity of MIR165/166 is localized to the periphery of the shoot meristem. (**A**) Expression of *MIR166Ap::GFPER* (green), PIN1-CFP (blue) and REV-2 × YPet (red) in the vegetative meristem (VM) at 3.5 DAS. (**B**) Longitudinal section of meristem shown in (**A**). (**C**) Expression of *MIR166Ap::GFPER* (green), PIN1-CFP (blue) and REV-2 × YPet (red) in the inflorescence meristem (IM). White arrow head marks the reestablishment of *MIR166Ap::GFPER* expression around the meristem after organ emergence. (**D** to **F**) Expression of REV-2 × YPet (red) alone (**D**), a MIR165/166 biosensor driven by the *UBQ10* promoter (green) alone (**E**) and both combined in the same IM (**F**). (**G**–**I**) Corresponding control for (**D** to **F**) where the MIR165/166 biosensor has been rendered insensitive to MIRNA activity. Bars represent 20 µm.

DOI: https://doi.org/10.7554/eLife.27421.005

## Expression patterns of REV and KAN1 in the shoot regulate leaf positioning and morphogenesis

To test whether boundaries between KAN1 and Class III HD-ZIP expression in the SAM can play an instructive role in positioning new organs and determining their subsequent dorsoventrality, we induced KAN1−2 × GFP expression ectopically at the center of the SAM using a pOp6/LhGR two-component (*Samalova et al., 2005*) system and the CLV3 promoter driving LhGR. After KAN1−2 × GFP induction, most seedlings initiated several new leaves before their growth stopped. Confocal imaging of seedlings five days after stratification (DAS) on dexamethasone (DEX) induction medium revealed that new organs, marked by high levels of PIN1-CFP expression, formed ectopically at the perimeter of an enlarged and irregular central domain of induced KAN1−2 × GFP expression, in which REV-2 × YPet expression had been repressed (*Figure 7A and B*). Although endogenous KAN1 was not monitored, ectopic KAN1−2 × GFP was only detected within or bordering organs during their initiation while REV-2 × YPet expression was often restricted during later developmental stages (*Figure 7C–H*). This indicates that patterns of KAN1 gene expression within the SAM can influence subsequent organ development. In particular, we noted that the distal margins of developing leaf primordia always correlated with boundaries of REV expression within the epidermis, even when REV expression was abnormally restricted (arrow heads in *Figure 7C–H*; *Video 1*). Several classes of phenotype, including leaves with an inverted orientation, could be distinguished at maturity (*Figure 7I–T*) that correlated with the patterns of REV-2 ×YPet expression

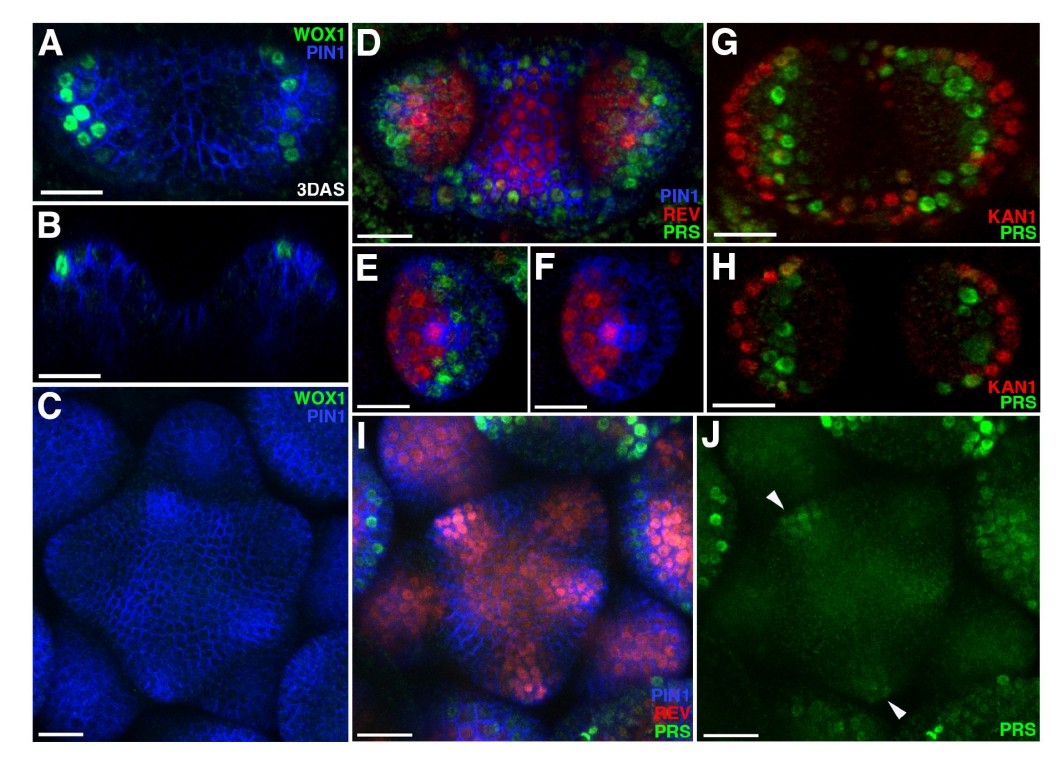

**Figure 4.** Expression patterns of 2 × GFP-WOX1 and PRS-2 ×GFP. Confocal projections showing PIN1-CFP (blue) and 2 × GFP-WOX1 (green) expression patterns in the vegetative meristem and leaves of seedlings at 3 DAS. (B) Longitudinal section of meristem shown in (A). (C) An inflorescence meristem image showing 2 × GFP-WOX1 is not expressed in the IM. (D) Confocal projection showing PIN1-CFP (blue), PRS-2 ×GFP (green) and REV-2 ×YPet (red) expression in the vegetative meristem and leaves at 3.5 DAS, where PRS-2 ×GFP is expressed surrounding the VM and along the leaf margins. (E and F) Cross sections of leaf on the right side in (D) showing the expression of PRS-2 × GFP in the middle domain of the leaf. (G and H) Confocal projection and cross section showing PRS-2 × GFP (green) and KAN1−2 × CFP (red) expression patterns in the vegetative meristem and leaves of seedlings at 3 DAS. (I and J) PRS-2 × GFP (green) is expressed in the young floral bracts in the IM, indicated with arrowhead in (J). Bar = 20 μm.

DOI: https://doi.org/10.7554/eLife.27421.006

observed during early development. These distinct morphological classes can be explained according to the configuration of the HD-ZIPIII-KAN boundaries at organ inception. Specifically, we infer that the number and orientation of boundaries within organ founder cells (as specified by high auxin levels) determines the configuration of later leaf marginal tissue (*Figure 7P–S*).

## Maximal auxin response is localized to HD-ZIPIII and KAN boundaries

The influence of dorsoventral gene expression on organogenesis suggests that dorsoventral boundaries may function generally to localize auxin response. In support of this proposal it has been reported that in the inflorescence meristem the auxin transcriptional marker DR5 is only responsive to auxin in the shoot periphery (*de Reuille et al., 2006*). We investigated DR5 expression (*Liao et al., 2015*) in the vegetative meristem by examining its expression both in the wild type and after 1-N-Naphthylphthalamic acid (NPA) treatment at 3 DAS. In mock-treated seedlings we found DR5 to be expressed at the locations of incipient primordia and at the distal tip of existing leaf primordia (*Figure 8A and B*). In contrast, NPA-treated seedlings expressed DR5 in a ring encircling the SAM prior to organ emergence and in the middle domain or dorsoventral boundary region of developing leaves (*Figure 8C and D*). Similar experiments with seedlings expressing the ratiometric R2D2 intracellular auxin reporter (*Liao et al., 2015*) revealed a generally broader distribution of auxin compared to the DR5 transcriptional response, especially after NPA treatment, where signal was not restricted to the meristem periphery or leaf tip (*Figure 8E–H*). These results indicate that like the

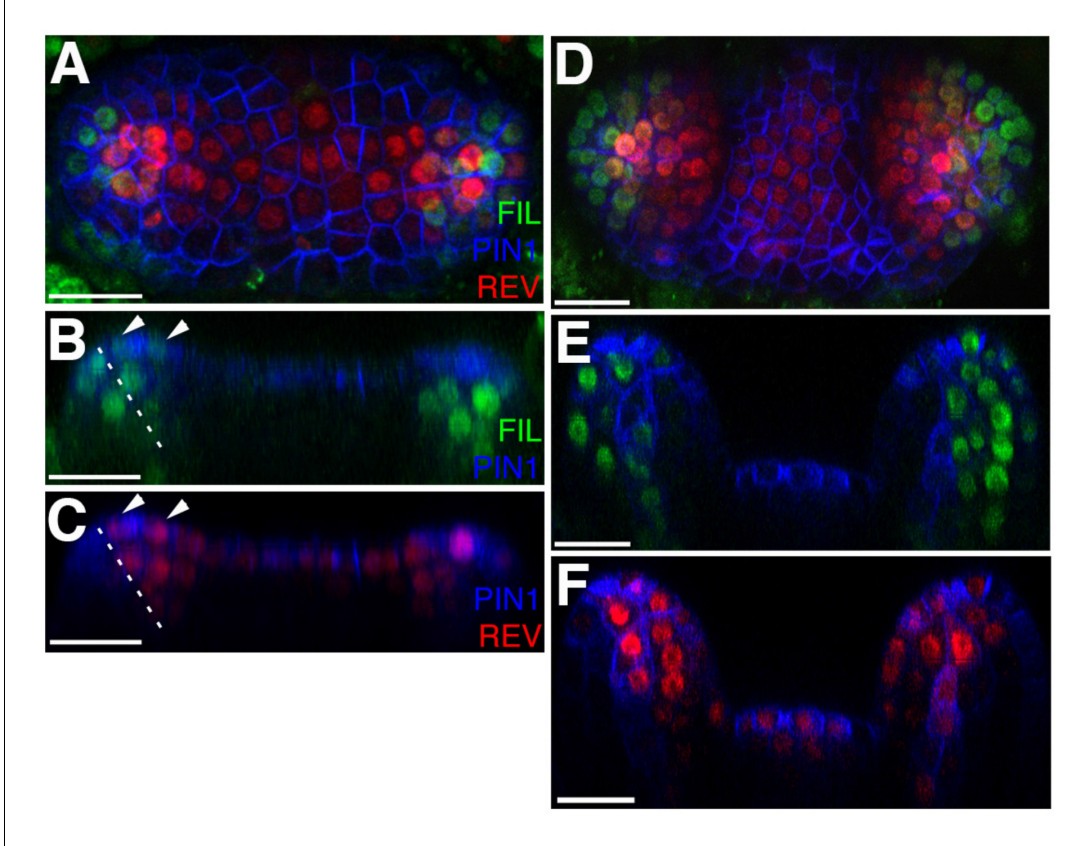

**Figure 5.** *FILp::dsREDN7* expression is broad during leaf initiation but is later excluded from dorsal tissues. (A–F) Confocal projections and reconstructed sections of seedlings expressing *FILp::dsREDN7* (green), REV-2 ×VENUS (red) and PIN1-CFP (blue). (A) Top view of seedling at 3DAS (B–C) Longitudinal section of seedling shown in (A). Dashed line shows dorsoventral axis of first leaf and arrowheads mark dorsal cells expressing both REV-2 ×VENUS and *FILp::dsREDN7*. (D) Seedlings at 3.5 DAS with *FILp::dsREDN7* expression more restricted to the developing ventral side of the leaf. (E–F) Longitudinal sections of seedling shown in (D) showing a more complementary pattern of *FILp::dsREDN7* relative to REV-2 ×VENUS compared to the earlier stage shown in (A) to (C). Scale bars represent 20 μm.

DOI: https://doi.org/10.7554/eLife.27421.007

inflorescence meristem, auxin transcriptional response outside the periphery of the vegetative meristem appears repressed. Since we also observed PRS expression localized to the meristem periphery (*Figure 4D*), we tested whether PRS as well as WOX1 are auxin inducible and found that both genes respond to auxin treatment within 12 and 15 hr respectively although their response was restricted to the boundary region (*Figure 8I–P*). Measuring transcript levels using qPCR indicated that the auxin response for both genes occurs at the transcriptional level (*Figure 8Q*).

These results not only reveal that general auxin response is maximized at the dorsoventral boundary but also, that genes already known to be expressed at the boundary are auxin responsive. Considering our results altogether and the finding that KAN1 represses PRS and WOX1 in ventral tissues (*Nakata et al., 2012*), we suggest a general scenario in which KAN and HD-ZIPIII genes repress auxin response where they are expressed, leaving a narrow domain of auxin responsive cells in between their expression domains.

We tested this proposal in silico by implementing a previous model for phyllotaxis (*Jönsson et al., 2006*; *Smith et al., 2006*), now supported experimentally (*Bayer et al., 2009*; *Bhatia et al., 2016*; *Heisler et al., 2010*), that incorporates the polarization of PIN1 towards cells with high intracellular auxin concentrations. By assuming that both KAN1 and REV repress auxin-induced transcription and by including a narrow region of cells located between the KAN1 and REV expression domains the model was able to self-organize the periodic formation of auxin maxima

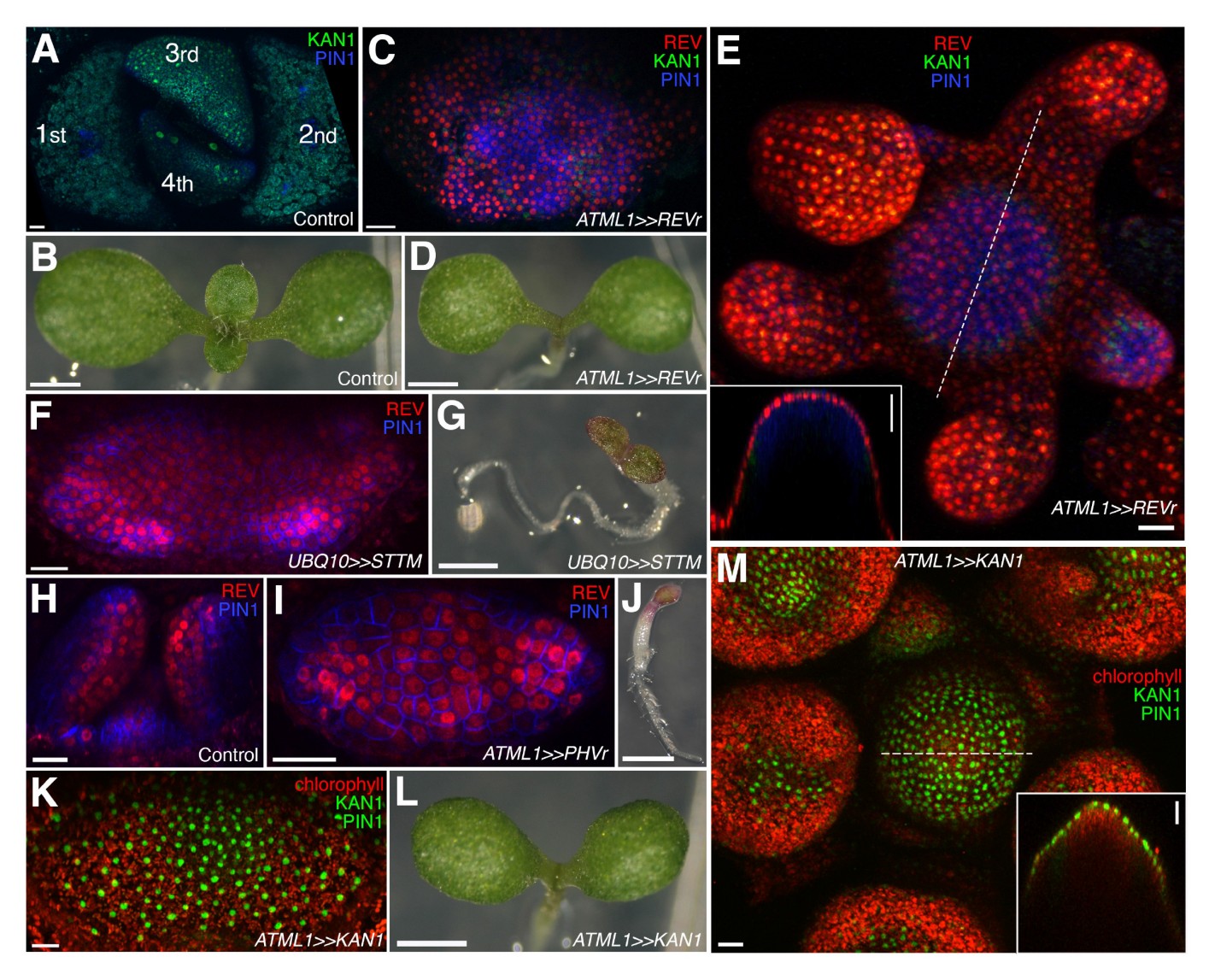

**Figure 6.** Organ initiation depends on the restriction of Class III HD-ZIP and KANADI expression in the shoot. (A) Confocal projection showing wild type control seedling at 7DAS viewed from above for comparison to (C) and (F). (B) Macroscopic view of control seedling at 7DAS for comparison to (D), (G) and (L). (C) Arrest of organogenesis after ectopic expression of REVr-2 × VENUS from the *ATML1* promoter in the vegetative meristem (7 DAS) after germination on DEX, KAN1−2 × GFP (green) expression is down regulated and could only be detected in a few cells in the sub-epidermis. Although PIN1-CFP (blue) expression is patchy, no leaves developed. (D) Macroscopic view of plant in (C). (E) Arrest of organogenesis after ectopic expression of REVr-2 ×VENUS (red) from the *ATML1* in the IM after 3 DEX treatments over 6 days. Note the absence of KAN1−2 × GFP signal. Inset shows longitudinal optical section of the meristem across the dashed line. (F) Seedlings at 7DAS showing similar phenotype to (C) after induction of a short tandem target mimic (STTM) designed to down regulate MIR165/166 activity. (G) Macroscopic view of plant in (F). (H–J) Ectopic expression of REV-2YPet (red) and arrest of organogenesis (PIN1-CFP in blue) in 4DAS seedling after induction of MIR165/166 resistant PHAVOLUTA. (H) Longitudinal view of un-induced control. Top view (I), and macroscopic view (J) of induced seedling showing arrest of organ development. (K and L) Confocal projection (K) and macroscopic view (L) of seedling at 7DAS after induction of KAN1-GFP (green) in the epidermis. No leaves have developed (autofluorescence shown in red). (M) Arrest of organogenesis after induction of KAN1-GFP (green) driven by the *ATML1* promoter in the IM after 3 DEX treatments over 6 days; autofluorescence (red). Inset shows longitudinal optical section of the meristem across the dashed line. Scale bars represent 20 μm in (A, C, E, F, H –I, K and M); 1 mm in (B, D, G, J and L).

DOI: https://doi.org/10.7554/eLife.27421.008

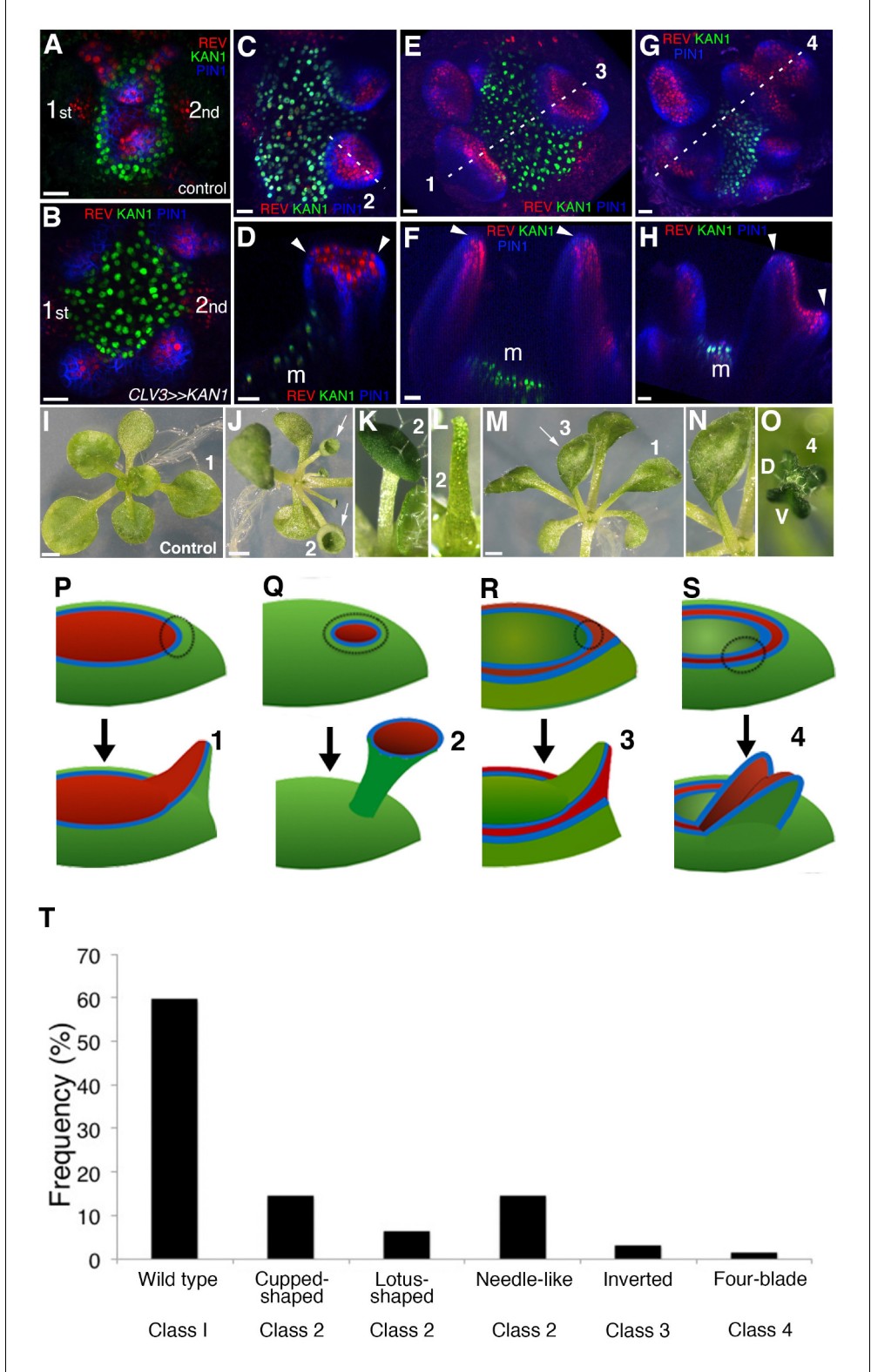

**Figure 7.** KANADI1 expression boundaries in the shoot specify organ position and orientation. (**A and B**) Confocal projections showing organ initiation marked by REV-2 × YPet (red) and PIN1-CFP (blue) at border of KAN1−2 × GFP expression (green) in wild type (**A**) and after induction of KAN1−2 × GFP using the *CLV3* promoter (**B**). Distance separating opposite organs was greater for induced (**B**) compared to control (**A**) (114.3 ± 3.3 μm, n = 19 vs 54.2 ± 1.0 μm n = 10 (mean ±SE, p<0.05, t-test)). (**C–H**) Confocal projections (**C, E and G**)
*Figure 7 continued on next page*

*Figure 7 continued*

and longitudinal reconstructions corresponding to dashed lines (**D, F and H** respectively) showing restricted REV-2 ×YPet expression (red) after ectopic KAN1−2 × GFP induction (green). Regions in which neither REV-2 ×YPet nor KAN1−2 × GFP signal was detected may potentially express endogenous KAN1, which was not monitored. Four main configurations of REV expression and morphology were observed (labeled 1 to 4). Class 1 organs (**E and F**) correspond to the wild type, Class 2 (**C and D**) express REV-2 ×YPet centrally, Class 3 (**E and F**) express REV-2 ×YPet in a reversed orientation and Class 4 (**G and H**) express REV-2 ×YPet centrally and laterally only. Correspondence between REV-2 ×YPet expression boundaries and leaf margins indicated by arrowheads in D, F and H; m indicates meristem. Gamma value changed to highlight PIN1-CFP expression (blue) in (**C**) to (**H**). (**I–O**) Examples of mature leaves corresponding to Classes 1 to 4, including the WT (**I**), cup-shaped (**J**), lotus-shaped (a variation of cup-shaped) (**K**), needle-like (a further decrease in extent of dorsal tissue compare to cup-shaped) (**L**), inverted (**M and N**) and four bladed (**O**). 'D and V' represent 'dorsal' and 'ventral' respectively in (**O**). (**P to S**) Diagrams summarizing proposed configurations of REV and KAN (green) gene expression in leaf founder cells (dashed circles) (upper diagram) leading to the observed phenotypic classes of leaf shape (numbered 1 to 4) (lower diagram) after induction of KAN1−2 × GFP using the *CLV3* promoter. (**P**) represents the wild type Class 1 configuration, (**Q**) represents Class 2, (**R**) represents Class 3 and (**S**) represents Class 4. (**T**) Frequency of seedlings exhibiting different leaf morphologies after ectopic induction of KAN1−2 × GFP expression in the CLV3 domain Class of phenotype corresponds to those indicated in (I to O). Scale bars = 20 μm in A to H; 1 mm in I, J and M.

DOI: https://doi.org/10.7554/eLife.27421.009

along the boundary as predicted, compared to a broader distribution of maxima when such boundaries are not included (***Figure 9A–D***; compare 9B to 9D).

## Auxin organises HD-ZIPIII and KAN expression in the shoot periphery

So far, our results indicate that both HD-ZIPIII and KAN1 suppress auxin-induced gene expression where they are expressed. However this does not exclude the possibility that auxin may play a role in patterning HD-ZIPIII and KAN expression. Indeed, auxin promotes the expression of several HD-ZIPIII genes in rice (***Itoh et al., 2008***) and in Arabidopsis, REV expression typically extends towards PIN1 polarity convergence patterns in the meristem periphery and disappears in axil regions where auxin is depleted (***Heisler et al., 2005***). To further investigate any potential regulation of REV and KAN expression by auxin we treated inflorescence meristems with synthetic auxin NAA(1-Naphthyl-acetic acid)as well as auxin transport inhibitor NPA (N-1-Naphthylthalamic acid) to prevent rapid auxin redistribution and examined the response of both REV and KAN1. We found that the region in between the REV and KAN1 domains narrowed due to a significant expansion of REV expression within 15 hr of treatment (***Figure 10A and B***). REV also became expressed in cells between the primordia and meristem where REV had presumably reduced its expression previously (***Figure 10C and D***). This was confirmed by following cells over time where we found that although KAN expression appeared static, more cells expressed REV (***Figure 10C–H***). A more subtle response was observed as early as 6 hr upon combined NAA and NPA application (***Figure 10—figure supplement 1***). Local application of IAA to *pin1-4* meristems also resulted in a local response by REV within 16 hr of auxin application (***Figure 10—figure supplement 2***). In contrast, exogenous auxin had little effect on a pCLV3:GFP-ER marker for central zone identity (***Figure 10—figure supplement 3***). Apart from changes in REV we noticed that over the course of 48 hr, KAN1 expression

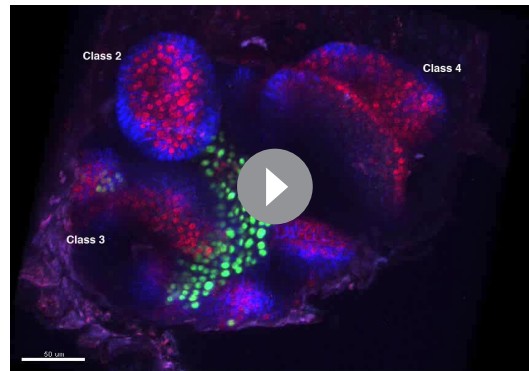

**Video 1.** Movie shows confocal 3D projection of a single vegetative seedling corresponding to that shown in ***Figure 7G and H***. Several leaf-like organs have formed on the boundary of ectopic KAN1 expression driven by the CLV3 promoter (green). These organs express REV (red) in restricted patterns corresponding to the three classes described in ***Figure 7*** (labeled Class 2, 3 and 4). Note developing leaf margins, marked by high PIN1 expression (blue) correlate with REV expression boundaries in the epidermis.

DOI: https://doi.org/10.7554/eLife.27421.010

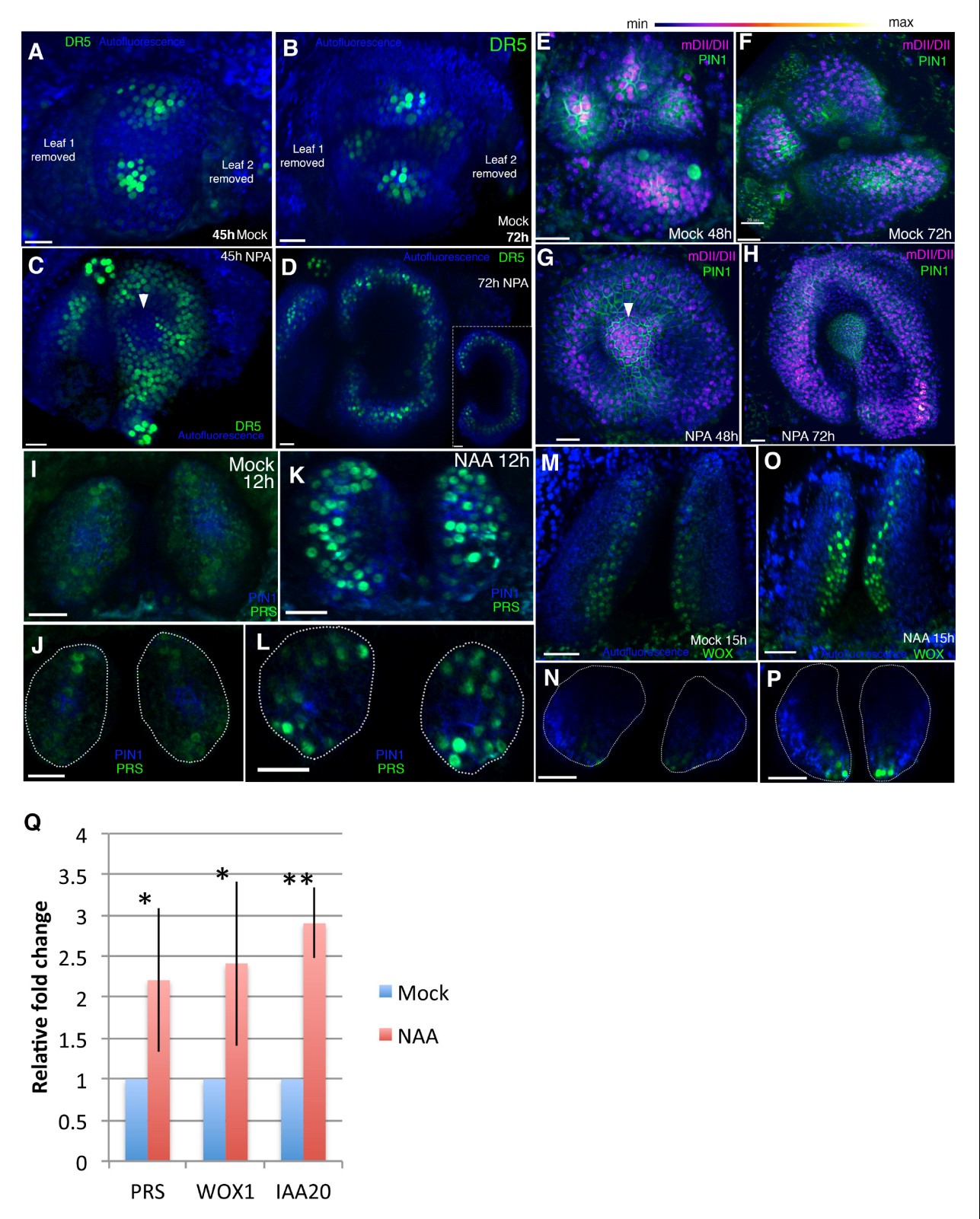

**Figure 8.** Auxin promotes PRS and WOX expression. (A–D) Response of pDR5V2–3 × VENUS-N7 (green) auxin transcriptional reporter to NPA in Arabidopsis seedlings treated at 3DAS (Days after Stratification). (A and B) Confocal projections 45 hr (A) and 72 hr (B) after treatment with mock solution. (C and D) Confocal projections 45 hr (C) and 72 hr (D) after treatment with 100 μM NPA solution (n = 5/5). White arrowhead in (C) marks absence of DR5 reporter in the center of the meristem. Inset in (D) shows transverse optical section through the ring-shaped organ showing most DR5

*Figure 8 continued on next page*

*Figure 8 continued*

expression localized in the center of the organ. (**E–H**) Expression and response of R2D2 (magenta) to auxin along with PIN1-GFP expression (green) in Arabidopsis seedlings treated at 3DAS. (**E and F**) Confocal projections 48 hr (**E**) and 72 hr (**F**) after treatment with mock solution. (**G and H**) Confocal projections 48 hr (**G**) and 72 hr (**H**) after treatment with 100 μM NPA solution (n = 4/4). White arrowhead in (**G**) marks the presence of auxin in the meristem center (compare with (**C**) which shows absence of auxin signaling in the meristem center). (**I–L**) Expression and response of pPRS:: PRS-2 ×GFP to auxin in Arabidopsis seedlings. Confocal projections and transverse optical slices of seedlings 4DAS showing of pPRS:: PRS-2 ×GFP expression (green) 12 hr after treatment with mock solution (**I and J**) and 5 mM NAA (**K and L**) (n = 5/5). (**M–P**) Expression and response of 2 × GFP WOX to auxin in Arabidopsis seedlings. (**M and N**) Confocal projections (**M and O**) and corresponding optical slices (**N and P**) of seedlings 4DAS showing pWOX1::2 × GFP-WOX1 expression (green) 12 hr after treatment with mock solution (**M and N**) and 5 mM NAA (**O and P**). Note WOX expression increases but does not expand beyond its regular expression domain upon auxin addition (n = 5/5). (**Q**) Q-PCR analysis of PRS, WOX1 and positive control IAA20 transcripts after 5 mM NAA or mock treatment on 4 days old wild-type (Ler) seedlings. *=p < 0.05, **=p < 0.001. Scale bars 20 μm (**A–I, K**); 15 μm (**J and L**); 30 μm (**M–P**).

DOI: https://doi.org/10.7554/eLife.27421.011

The following source data is available for figure 8:

**Source data 1.** Statistical details of qRT-PCR.

DOI: https://doi.org/10.7554/eLife.27421.012

did not appear as expected in between the meristem and floral bracts of stages i1 and i2 (but not P1) at the time of treatment (*Figure 10I and J*), compared to mock treated plants (*Figure 10K and H*), where KAN expression appeared adjacent to the primordia within 24 hr. In vegetative meristems this was more obvious where contiguous REV expression often appeared between the meristem and developing leaves where KAN1 expression would normally be expressed (*Figure 11A–H*; *Video 2*). These results indicate exogenous auxin promotes HD-ZIPIII expression locally and that exogenous auxin can prevent the establishment of KAN1 expression if applied at stages i1 and i2 of organ development.

To test whether REV or KAN1 expression depend on endogenous auxin for their patterns of expression we treated triple marker plants with the TIR1 antagonist auxinole (*Hayashi et al., 2012*) and found that after 18 hr of treatment KAN expression had expanded slightly toward the meristem center (*Figure 12A and B*; *Figure 12—figure supplement 1A–D*). As auxinole competes with endogenous auxin for the TIR1 binding pocket, we attempted to reduce endogenous auxin signaling further by simultaneously treating the meristems with yucasin (*Nishimura et al., 2014*) and kyn (*He et al., 2011*) to block auxin synthesis. This combination of treatments led to an arrest of organogenesis and 'pin' phenotype as well as a much more significant expansion and restriction of KAN1 and REV expression centrally, respectively (*Figure 12C and D*; *Figure 12—figure supplement 1E–H Figure 12—figure supplement 2*). KAN expression not only expanded into those cells originally located between the REV and KAN1 domains but also cells that had previously been expressing REV at high levels. However KAN1 expression remained excluded from the central most cells of the shoot and established floral bracts and leaf primordia, where REV expression still remained (*Figure 12D*; *Figure 12—figure supplement 2*. Extending the drug treatment time did not lead to a further change in expression. Monitoring the ratiometric auxin sensor R2D2 in response to a similar combined inhibitor treatment confirmed a strong decrease in overall intracellular auxin levels, including in the central meristem region and maturing organs (*Figure 12E and F*), indicating that auxin independent mechanisms regulate the expression patterns of these genes in the central zone and in organs after their initiation. In contrast to REV and KAN1, auxin depletion did not influence size of the central zone, as marked by the expression of pCLV3:GFP-ER (*Figure 12—figure supplement 3*).

Overall these data indicate that dynamic changes in auxin levels specifically within the meristem periphery play a central role in regulating dorsal and ventral identities. While high levels of auxin act to both trigger and maintain HD-ZIPIII expression, this only occurs in cells not already expressing KAN1. Such high levels of auxin also prevent the acropetal expansion of KAN1 expression, not only in cells within primordia but also in regions between the primordia and meristem. As our auxin treatments only prevented KAN1 expression from establishing around primordia at stages, i1 and i2 and not around P1, which corresponds to a stage when auxin depletion has already occurred (*Heisler et al., 2005*), our data also suggest that once cells are exposed to low auxin levels, their fate becomes determined, i.e subsequent auxin exposure cannot reverse their trajectory towards ventral or peripheral cell identity. Outside the meristem periphery, both in the meristem central

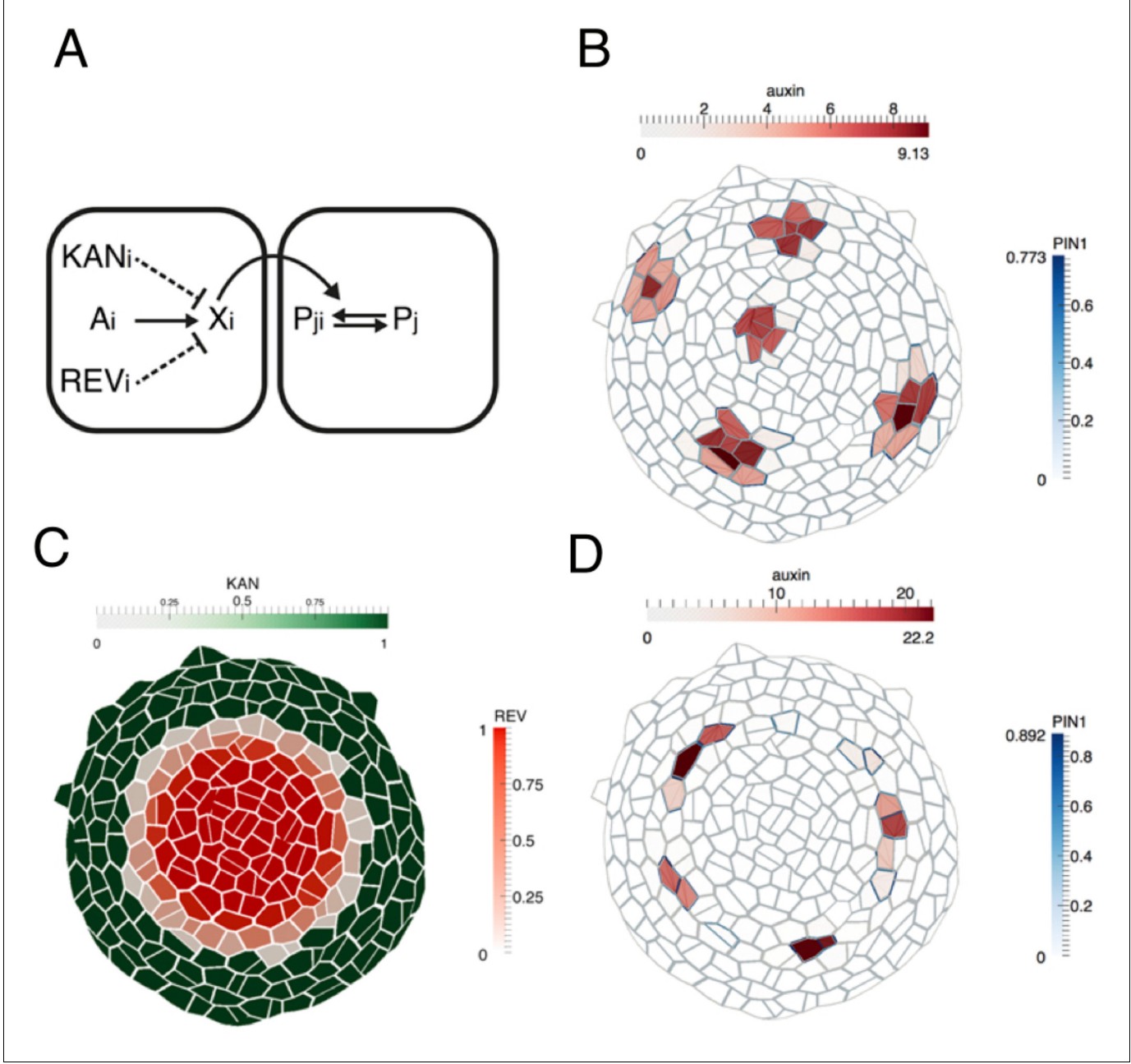

**Figure 9.** Computational model illustrating how dorsoventral gene expression boundaries may restrict phyllotactic patterning to the SAM peripheral zone. (A) Illustration of model interactions. Auxin is transported passively and actively via PIN1 between cells. PIN1 is polarized towards cells with high auxin, via a signaling pathway represented by X (previously suggested to be realized by increased stresses in the neighboring cells due to changes in mechanical wall properties (*Heisler et al., 2010*). (B) As shown previously (*Heisler et al., 2010*; *Jönsson et al., 2006*; *Smith et al., 2006*), peaks of auxin are formed spontaneously. (C) A pattern of KANADI (green) and REVOLUTA (red) is added to the template with a boundary domain in between in which REV expression is low or absent and KAN1 expression is absent. (D) If KANADI and REVOLUTA decrease the signal X in cells where they are expressed (dashed interactions in A), the formation of auxin peaks is restricted to the boundary.

DOI: https://doi.org/10.7554/eLife.27421.013

zone and within more established organs, REV and KAN expression appears comparatively auxin insensitive.

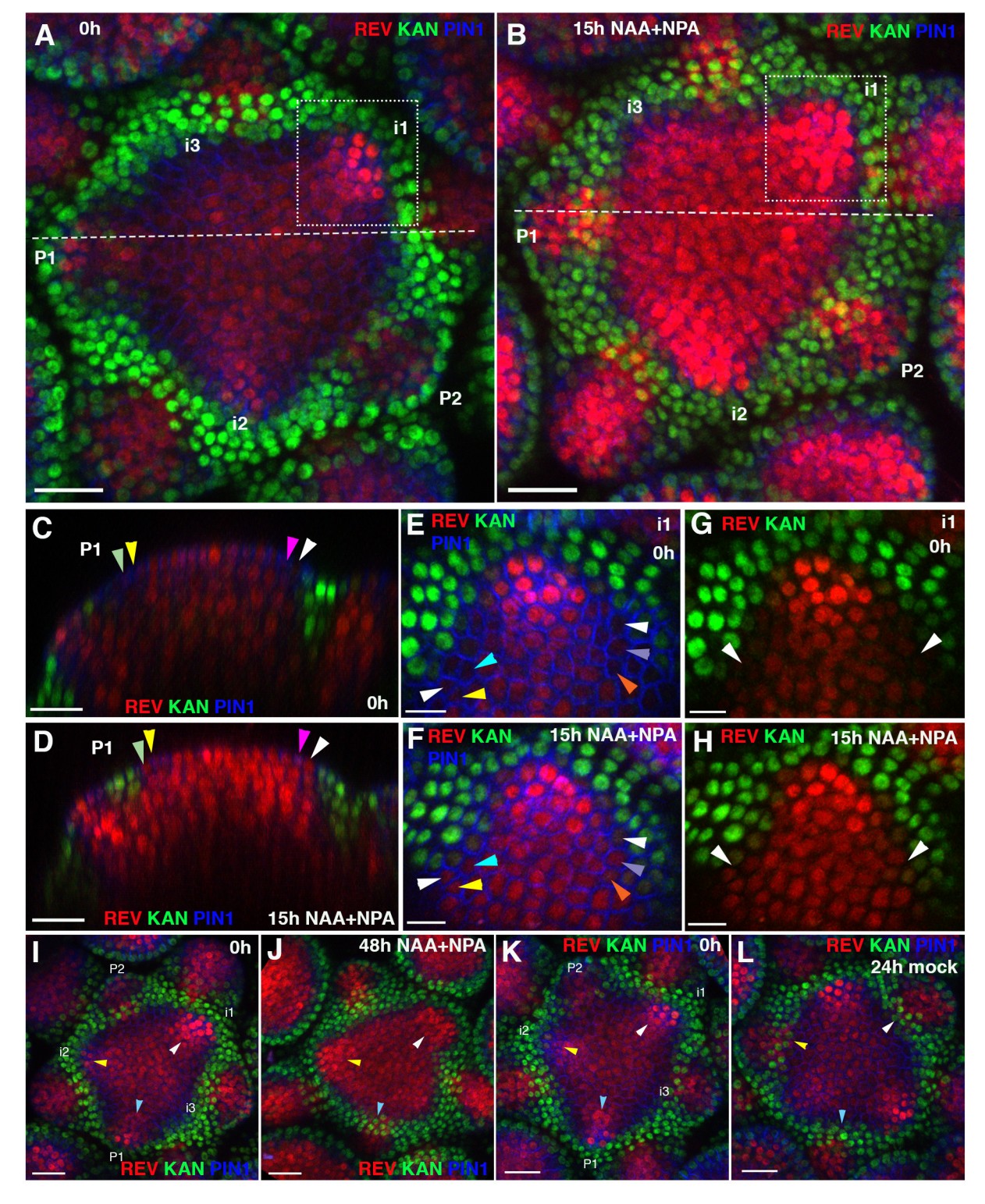

**Figure 10.** Effect of auxin on the expression patterns of REV and KAN1 in the inflorescence meristem. (**A and B**) Confocal projections of the IMs showing expression pattern of REV-2 × YPet (red), KAN1–2 × GFP (green) and PIN1-CFP (blue) before (**A**) and 15 hr after the combined application of 5 mM NAA and 100 µM NPA (**B**). Primordium (**P**) and incipient primordium (**i**) stages are numbered from i3-P2 based on convention described in (*Heisler et al., 2005*). Note up regulation and expansion of REV expression 15 hr after the combined application of NAA and NPA (n = 6). (**C and D**) Longitudinal optical sections along the dashed white lines in (**A**) and (**B**) respectively. Note the presence of REV expression in the epidermal cells

*Figure 10 continued on next page*

*Figure 10 continued*

marked by arrowheads in (**D**) and a corresponding absence or weak level of expression in (**C**). Similar colored arrowheads mark the same cells tracked over 15 hr. (**E and F**) Magnified views of the surface of i1, outlined by dotted rectangles in (**A and B**) showing expression pattern of REV-2 ×YPet (red), KAN1−2 × GFP (green) and PIN1-CFP (blue) before (**E**) and 15 hr after the combined application of 5 mM NAA and 100 µM NPA (**F**). Note the presence of REV in the cells marked by arrowheads in (**F**) and their absence in (**E**). Similar colored arrowheads mark the same cells tracked over 15 hr in (**E and F**). (**G and H**) Same as (**E and F**) but showing REV and KAN1 expression only. Note the presence of a gap between REV and KAN1 expression in (**G**) but its absence in (**H**), marked by white arrowheads in (**G and H**). (**I–L**) Confocal projections of IMs showing expression of REV-2 ×YPet (red), KAN1−2 × GFP (green) and PIN1-CFP (blue) before (**I**) and 48 hr after the combined application of 5 mM NAA and 100 µM NPA (**J**), and before (**K**) and 24 hr after treatment with mock solution (**L**). Initially for both control and treated meristems, KAN1 expression is absent between the meristem center and P1 (blue arrowheads), i1 (white arrowheads) and i2 (yellow arrowheads) (**I and K**). Under mock treatment, KAN1 expression appears in all three corresponding regions (marked by the same arrowheads) 24 hr later (**L**). However for the meristem treated with NAA and NPA, KAN1 expression is absent in regions previously corresponding to i1 (white arrowheads) and i2 (yellow arrowheads) but not P1 (blue arrowheads), even after 48 hr (**J**). Scale bars 20 µm (**A–D, I–L**) and 10 µm (**E–H**).

DOI: https://doi.org/10.7554/eLife.27421.014

The following figure supplements are available for figure 10:

**Figure supplement 1.** REV expression starts to expand within 6 hr of auxin treatment.

DOI: https://doi.org/10.7554/eLife.27421.015

**Figure supplement 2.** Auxin modulates the expression of REV and KAN1 locally.

DOI: https://doi.org/10.7554/eLife.27421.016

**Figure supplement 3.** Auxin application does not expands CLAVATA3 expression and hence, the central zone of the meristem.

DOI: https://doi.org/10.7554/eLife.27421.017

## Wounding induces KAN and represses REV expression in an auxin depletion-dependent manner

Since auxin is required to maintain REV at the expense of KAN expression in the meristem periphery and wounding causes repolarization of PIN1 away from wound sites (*Heisler et al., 2010*) resulting in auxin depletion (*Landrein et al., 2015*), we investigated whether wounding results in ectopic KAN1 and reductions in REV expression. Firstly, using a pulsed IR laser to ablate cells adjacent to young organ primordia we confirmed that auxin levels decrease in the vicinity of wounds by monitoring the expression of the R2D2 ratiometric marker (*Figure 13A and B*). Next, we monitored REV, KAN and PIN1 expression in response to such wounds. We found that KAN1 became expressed in cells adjacent to the wound on either side, regardless of wound orientation with respect to the SAM (*Figure 13C and D*; *Figure 13—figure supplement 1A–H*). Such a response argues against the possibility that ectopic KAN is the result of interruption of a signal emanating from the meristem that promotes dorsal and represses ventral identity (*Sussex, 1955*). Instead, it supports the proposal that KAN1 expression is promoted in the vicinity of wounds in general, possibly due to low auxin levels. To test this hypothesis, we repeated these experiments while treating the wounded meristems with combinations of NAA and NPA over a 48 hr period and found that the induction of KAN1 expression around wounds could be completely eliminated if NAA and NPA were combined (*Figure 13E–G*; *Figure 13—figure supplement 1I–L*). Although a similar response to wounding was found to occur in vegetative meristems, the wound response typically involved a more substantial reorganization of meristem structure, possibly due to the small size of the vegetative meristem relative to the wounds (*Figure 13—figure supplement 1M–O*). When new leaves subsequently formed, they were properly oriented with respect to the new meristem organization.

## Discussion

In this study we shed new light on a long-standing question regarding leaf dorsoventrality in plants – when and how is it established? The early work of Sussex, based on histological analysis and wounding experiments, suggested that initiating leaf primordia require an inductive signal from meristem tissues to specify dorsal cell fate (*Sussex, 1955*). This proposal has been further supported by a more recent study in tomato (*Reinhardt et al., 2005*). In contrast, other workers in the field have claimed that dorsoventrality arises directly from radial patterning of the shoot (*Husbands et al., 2009*; *Kerstetter et al., 2001*), with some authors also linking the organogenic capacity of the SAM peripheral zone to the meristematic nature of leaf margins (*Hagemann and Gleissberg, 1996*) and

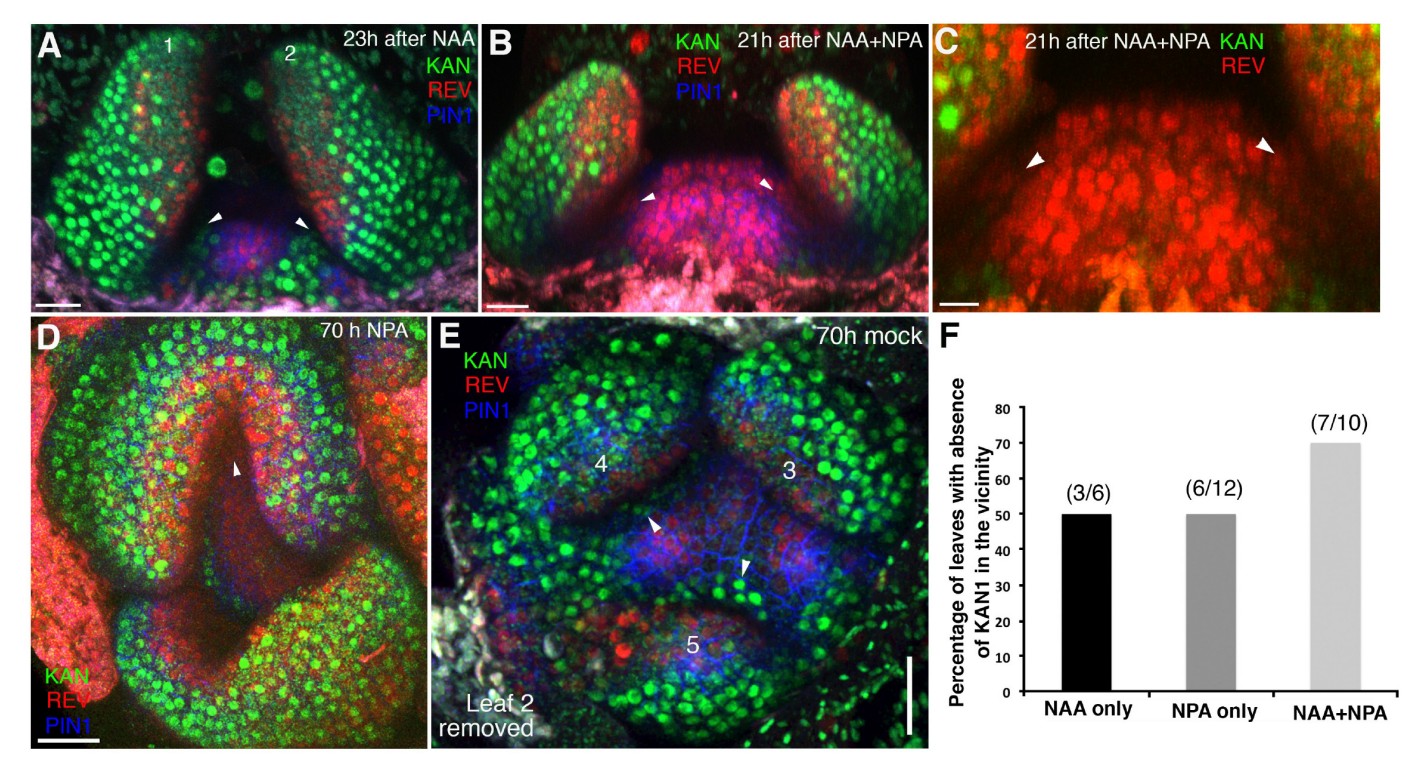

**Figure 11.** High auxin levels prevent new KAN1 expression in the leaf axils. (A–E) Confocal projection of the VM showing expression pattern of REV-2 × YPet (red), KAN1–2 × GFP (green) and PIN1-CFP (blue) 23 hr after the treatment with 5 mM NAA (A), 21 hr after combined treatment with NAA and NPA (B and C); panel (C) showing close-up view of the meristem in (B) with REV (red) and KAN (green) expression only, 70 hr after treatment with NPA alone (D) and 70 hr after treatment with mock (E). White arrowheads mark the presence of KAN1 expression in the cells adjacent to the grown out leaves in (A) and (E) and absence in (B–D). Note that REV expression (red) expanded towards leaves axils in (B–D). However also note that REV expression appears faint in the leaf axils in (B and C) compared to (D). This is due to the reason that the combined application of NAA and NPA resulted in the leaves growing at an acute angle to the meristem, which caused shading and made it difficult to reach leaf axils while imaging. (F) A bar graph showing percentages of leaves lacking KAN1 expression in their vicinity upon treatment with NAA, NPA and NAA + NPA. Scale bars 20 μm (A and B), 10 μm in (C) and 30 μm (D and E).
DOI: https://doi.org/10.7554/eLife.27421.018

dorsoventral boundaries (*Koch and Meinhardt, 1994*). Our results support elements of these latter proposals by revealing that organs are pre-patterned by domains of KAN and HD-ZIPIII expression not only in terms of their differentiation but also their position and morphology. The effect of ectopic KAN1 expression at the centre of the shoot on subsequent leaf positioning and growth is particularly striking and indicates that the spatial arrangement of HD-ZIPIII and KAN expression present within organ founder cells is incorporated into organs as they initiate and directs patterns of morphogenesis (*Figure 7P–S*; *Figure 14*). However, the exact configuration of this pre-pattern, including its propagation into developing organs, is dynamic and depends on changing auxin levels, although this regulation is unlikely to be direct. While high auxin levels promote HD-ZIPIII and represses KAN1 this is only true for cells not already expressing KAN1. Hence, at sites of incipient organ formation, where auxin is concentrated in between the two expression domains, HD-ZIPIII expression is promoted and can expand peripherally, but only in cells adaxial to where KAN is expressed. At the same time, this high level of auxin prevents KAN expression from expanding acropetally into HD-ZIPIII expressing cells, causing the boundary to remain relatively static with respect to the underlying cells (*Figure 14A*). However if auxin levels drop, for instance when PIN1 reverses polarity away from cells between the meristem and primordium (*Heisler et al., 2005*; *Qi et al., 2014*), HD-ZIPIII expression decreases and KAN expression takes its place (*Figure 14B and C*). Hence high levels of auxin 'lock-in' the spatial arrangement of HD-ZIPIII and KAN expression within organ founder cells at organ inception until a later stage when the pattern becomes auxin

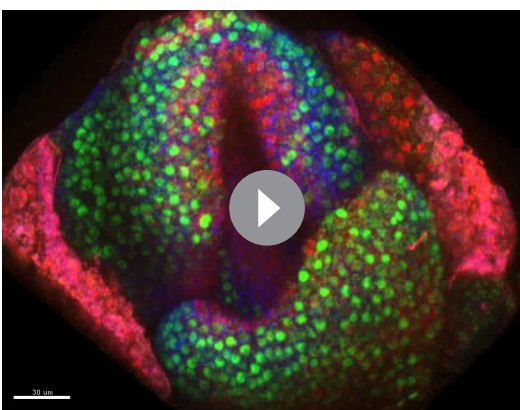

**Video 2.** Confocal projection of an Arabidopsis seedling 70 hr after treatment with NPA showing view of the vegetative meristem from above with channels for PIN1-CFP (blue), REV-2x YPet (red) and KAN1-2xGFP (green) alternating (also shown as a snapshot in *Figure 11C*). Note the absence of contiguous KAN1 expression in between the meristem and older leaves potentially due to auxin build up in the absence of its regular transport.

DOI: https://doi.org/10.7554/eLife.27421.019

independent. In between primordia, where auxin levels are lower, cells continuously transition from expressing REV in the central zone (where HD-ZIPIII expression appears auxin insensitive) to expressing KAN1 as they are displaced peripherally, as also indicated by the expression patterns of these genes in *pin1* meristems (*Figure 10—figure supplement 2A,E and I*).

Overall these results imply that the dorsoventral patterning of organs results from a four-step process: (i) signals during embryonic development establish concentric patterns of Class III HD-ZIP and KAN gene expression; (ii) the boundary between these domains helps to define sites for auxin-dependent organogenesis i.e. the meristem peripheral zone; (iii) As organs form, dynamic auxin levels modulate the patterns of HD-ZIPIII and KAN expression, thereby helping to position new organs and establish their dorsoventrality; (iv) patterns of HD-ZIPIII and KAN gene expression within organs are stabilized and dictate future patterns of morphogenesis.

The regulation of HD-ZIPIII and KAN expression by auxin is not only relevant to understanding wild type organ development but also for understanding the reorganization of tissue types in response to wounds. Wounds in the meristem outer cell layer specifically alter cell polarities such that PIN1 becomes polarized away from wounds in adjacent cells (*Heisler et al., 2010*), leading to auxin depletion (*Landrein et al., 2015*). Consistent with this, we observe KAN1 to be upregulated and REV to be repressed around wounds. Furthermore, such changes can be completely blocked by auxin application. Altogether then, our results strongly indicate that wounds re-pattern peripheral cell identities due to the depletion of auxin in their vicinity. Specifically, we propose that auxin depletion triggers a local commitment to peripheral cell fate (which includes expressing KAN1), thereby providing a parsimonious explanation for the wound-associated disruptions to leaf polarity observed by Sussex and others (*Sussex, 1955*). However further experiments using the vegetative meristems of plants with larger meristems will be needed to confirm this proposal since we were not successful in altering the dorsoventrality of Arabidopsis leaves by wounding and therefore could not test whether auxin treatment could prevent such alterations.

Our conclusion that auxin promotes dorsal cell fate while inhibiting ventral cell fate contrasts with those of another study reporting that transient high levels of auxin inhibit establishment or maintenance of dorsal fate and/or promote establishment or maintenance of ventral cell fate (*Qi et al., 2014*). These findings were based on auxin application experiments in tomato that resulted in the ventralization of leaves as well as the observation that Arabidopsis *pin1* and *pid rev* double mutants produce trumpet or rod-shaped shaped leaves. Although our results cannot easily explain the tomato data, since we find that high auxin levels are required to maintain REV expression during organ establishment, we would expect that in *pin* and *pid* mutants, lower auxin levels may well result in lower expression levels of REV and possibly other Class III HD-ZIPs, leading to leaf ventralization as reported in these mutant backgrounds (*Qi et al., 2014*). Further auxin application experiments in tomato that include an analysis of gene expression and auxin level changes may clarify the relationship between auxin and dorsoventral patterning in tomato compared to Arabidopsis.

How do DV boundaries control organ position and shape? Both our data as well as data from previous studies suggest this is through the regulation of auxin perception. For instance, organ initiation requires high auxin levels but auxin can only trigger organogenesis in the peripheral zone (*Reinhardt et al., 2000*) where the HD-ZIPIII/KAN boundary occurs. A similar relationship holds for leaves where auxin application results in growth but only from the leaf margins, again corresponding to a DV boundary (*Koenig et al., 2009*). We further show that the localization of auxin response to

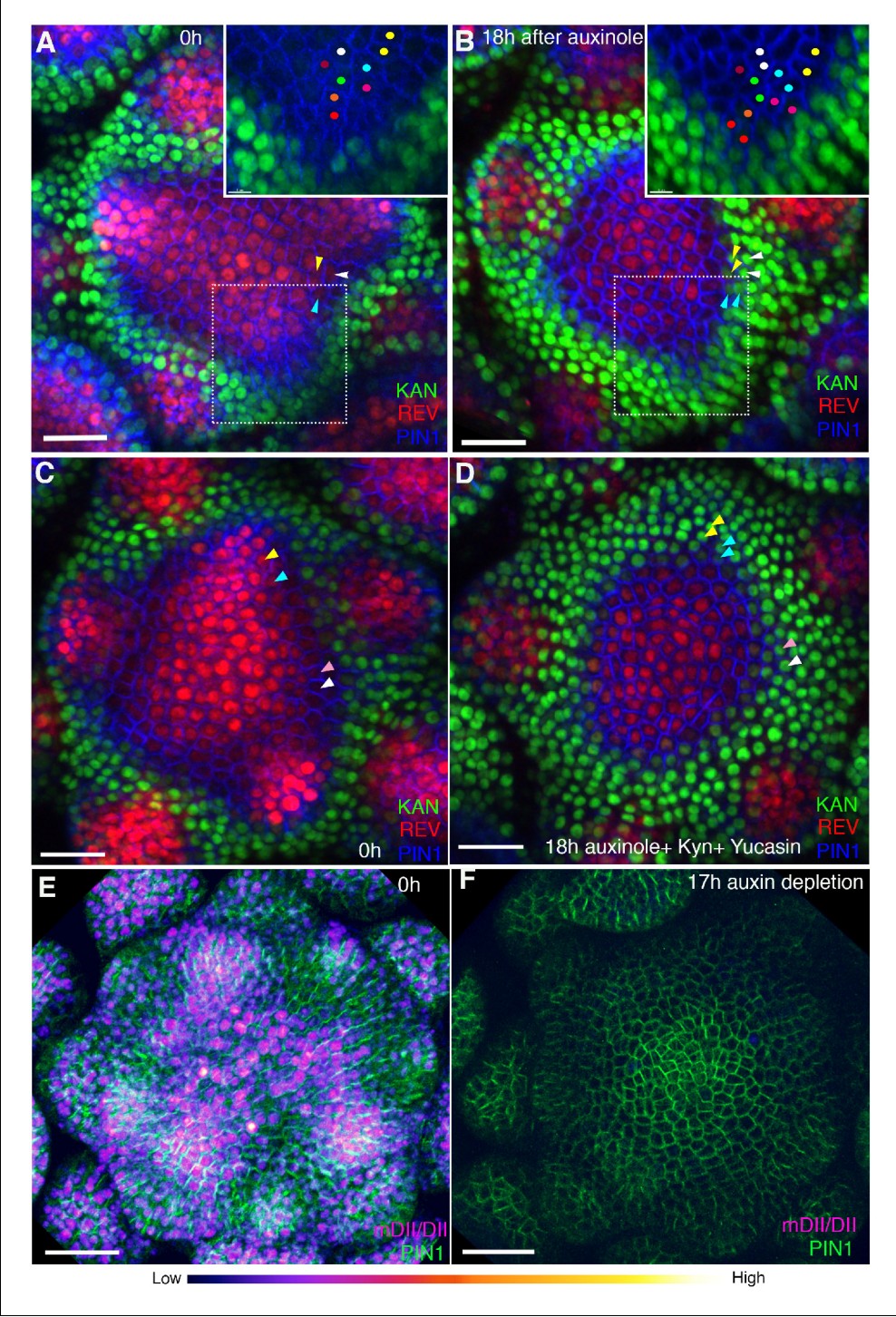

**Figure 12.** Auxin depletion alters boundary position. (**A and B**) Confocal projections of the IMs showing expression pattern of REV-2 × YPet (red), KAN1−2 × GFP (green) and PIN1-CFP (blue) before (**A**) and 18 hr (**B**) after the application of 100 μM auxinole. Inset shows close-up of the primordium outlined with the dotted rectangle. Similar colored dots mark the same cells at 0 hr and 18 hr time-points. Note the presence of KAN1 expression in the proximity of the cells marked with colored dots in the inset in (**A**) and its absence in the inset in (**B**). Similar color arrowheads in (**A and B**) mark the same cells that showed REV expression at 0 hr but KAN1 expression at 18 hr after treatment with auxinole. (**C and D**) Confocal projections of the IMs showing expression pattern of REV-2 × YPet (red), KAN1−2 × GFP (green) and PIN1-CFP (blue) before (**C**) and 18 hr (**D**) after the combined application of 100 μM auxinole, 100 μM KYN and 100 μM Yucasin (auxin depleting drugs). Note KAN-

*Figure 12 continued on next page*

*Figure 12 continued*

2 ×GFP expression has expanded centrally at the expense of REV-2 × YPet expression (compare the cells marked by arrowheads in (C) with (D), similar colored arrowheads mark the same cells tracked over 18 hr) (n = 6/6). (E and F) Confocal projections of the IMs indicating the predicted auxin distribution (magenta) based on R2D2 expression along with PIN1-GFP expression (green) before (E) and 17 hr after the combined application of 100 μM auxinole, 100 μM kyn and 100 μM yucasin (auxin depleting drugs) (F). Note lack of detectable auxin based on R2D2 expression in (F) compared to (E) after the combined drug application (n = 3/4). Scale bars 20 μm (A–D), 30 μm (E and F).

DOI: https://doi.org/10.7554/eLife.27421.020

The following figure supplements are available for figure 12:

**Figure supplement 1.** Auxin depletion results in KAN expression at the expense of REV.

DOI: https://doi.org/10.7554/eLife.27421.021

**Figure supplement 2.** Auxin depletion alters dorsoventral gene expression in the vegetative meristems.

DOI: https://doi.org/10.7554/eLife.27421.022

**Figure supplement 3.** Auxin depletion does not results in the shrinking of the central zone of the meristem.

DOI: https://doi.org/10.7554/eLife.27421.023

DV boundaries applies generally, i.e. DR5 expression appears higher at such boundaries compared to the broader predicted auxin distribution, as marked by R2D2. How is this restriction achieved? Imaging data reveals that the locations of PIN1 polarity convergences correspond to a region of cells in between the expression domains of HD-ZIPIII and KAN where their expression is low or absent. Given our results as well as genetic and molecular data indicating that both the HD-ZIPIII and KAN genes repress auxin activity (*Huang et al., 2014*; *Merelo et al., 2013*; *Müller et al., 2016*; *Zhang et al., 2017*), we propose that this localized absence of expression results in a local de-repression of auxin-induced transcription. Such localized auxin activity has recently been shown to orient cell polarity, including microtubule orientations, in a non-cell autonomous manner (*Bhatia et al., 2016*). Hence, we suggest that the proposed ability of dorsoventral boundaries to act as organizers analogous to those of the Drosophila wing (*Diaz-Benjumea and Cohen, 1993*; *Waites and Hudson, 1995*) rests in-part on their ability to orient cell polarity non-cell autonomously by localizing auxin activity (*Figure 14D*). The resulting growth may occur either in a periodic fashion along the boundary typical of phyllotaxis and complex leaves, or in a more continuous manner typical of simple leaves, depending on the strength of auxin transport or other modifications to the cell polarity feedback system (*Bilsborough et al., 2011*; *Koenig et al., 2009*). Investigating the dynamic consequences of directly juxtaposing REV and KAN expression should enable the testing of these hypotheses as well as lead to a better understanding of how auxin-responsive boundary domains are first established.

## Materials and methods

### Plant material

Plants were grown on soil at 22°C in continuous light-conditions and cultivated either on soil or on GM medium (1% sucrose, 1 × Murashige and Skoog basal salt mixture, 0.05% MES 2-(MN-morpholino)-ethane sulfonic acid, 0.8% Bacto Agar, 1 % MS vitamins, pH 5.7 with 1 M potassium hydroxide solution).

### Construction of transgenes

Multiply transgenic lines were generated by Agrobacterium-mediated transformation into stable transgenic lines or by genetic crossing. The *FILp::dsREDN7* and *PIN1p::PIN1-GFP* transgenes have been described elsewhere (*Heisler et al., 2005*). *pREV::REV-2×VENUS* in the T-DNA vector *pMLBART* (*Gleave, 1992*) is a modification of *pREV::REV-VENUS* (*Heisler et al., 2005*) that contains a translational fusion to two tandem copies of the fluorescent protein VENUS (*Nagai et al., 2002*). *REVp::REV-2×Ypet* containing a C-terminal fusion to the 2 × Ypet (*Nguyen and Daugherty, 2005*) in *pMOA36* T-DNA (*Barrell and Conner, 2006*) was transformed into a *PINp::PIN1-CFP* line (*Gordon et al., 2007b*). The *KAN1p::KAN1−2 × GFP* transgene in pMOA34 T-DNA was created by

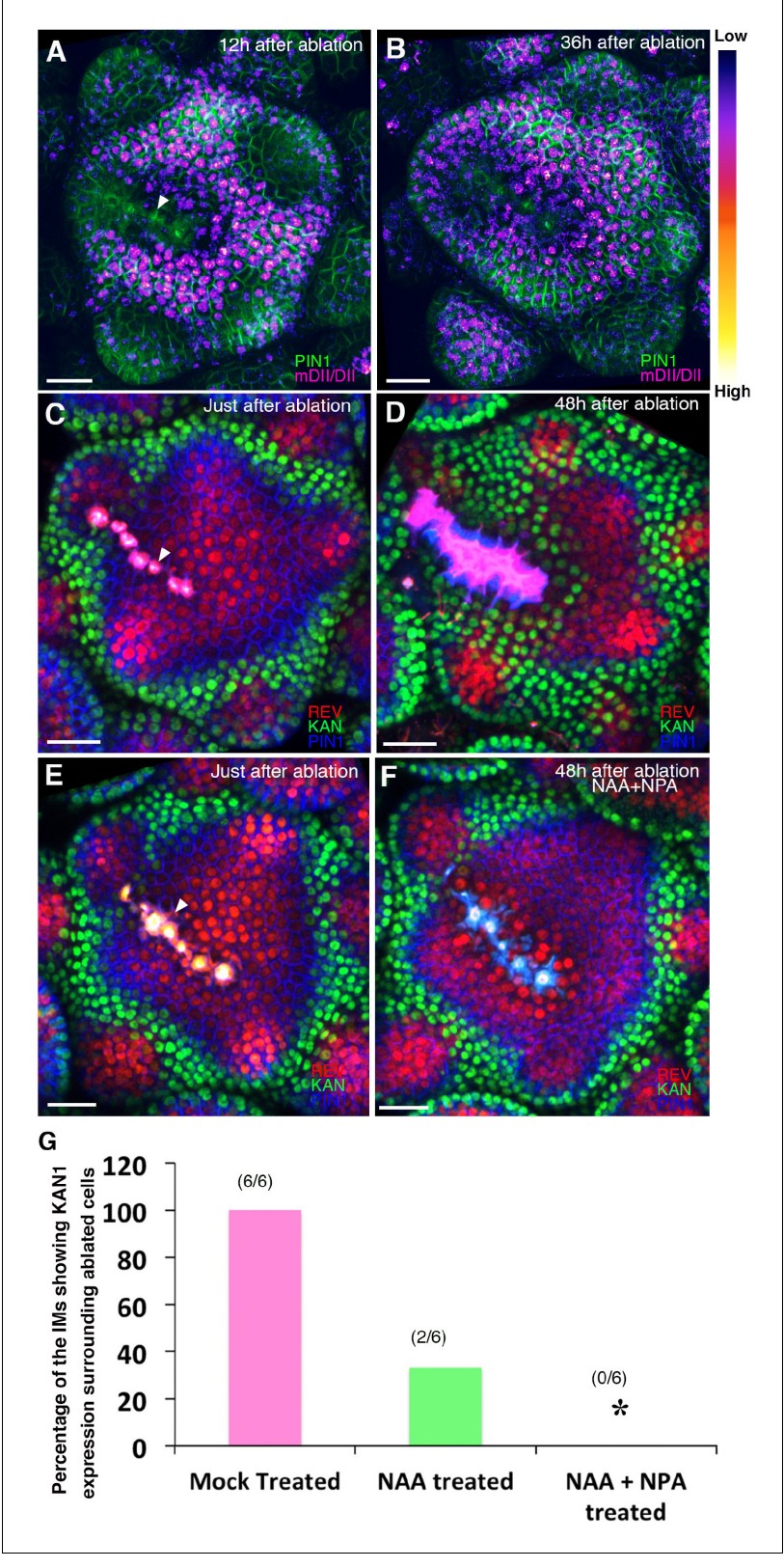

**Figure 13.** Wounding induces KANADI1 expression in response to low auxin. (**A and B**) Confocal projections of IMs showing predicted auxin distribution (magenta) based on R2D2 expression 12 hr (**A**) and 36 hr (**B**) after wounding. Note low predicted auxin levels in the cells surrounding the ablated cells. (**C and D**) Confocal projections of the IMs showing expression pattern of REV-2 × YPet (red), KAN1−2 × GFP (green) and PIN1-CFP

*Figure 13 continued on next page*

*Figure 13 continued*
(blue) immediately after ablation (ablated cells indicated a white arrowhead) (**C**) and 48 hr after (**D**). Note KAN1 expression (green) has completely surrounded the wounded cells 48 hr after the ablation (**D**) compared to (**C**). (**E** and **F**) Confocal projections of the IMs showing expression pattern of REV-2 × YPet (red), KAN1−2 × GFP (green) and PIN1-CFP (blue) immediately after ablation (ablated cells are marked by white arrowhead) (**E**) and 48 hr after ablation and combined NAA and NPA application (**F**). Note absence of KAN1 expression (green) surrounding the wound when wounding is accompanied by the exogenous addition of auxin and NPA (compare to (**D**)). (**G**) Quantification of wounding induced ectopic KAN1 expression upon mock treatment (n = 6/6)), NAA application (n = 2/6)) and NAA + NPA combination (n = 0/6) application on the Arabidopsis IMs expressing REV-2 ×YPet, KAN1−2 × GFP and PIN1-CFP. Scale bars 30 μm (**A–F**).
DOI: https://doi.org/10.7554/eLife.27421.024
The following figure supplement is available for figure 13:

**Figure supplement 1.** Wounding induces ectopic KAN expression in the inflorescence and vegetative meristems.
DOI: https://doi.org/10.7554/eLife.27421.025

amplifying 8.7 kb of *KAN1* (At5g16560) genomic sequences with primers KAN1g F and KAN1g R (*Supplementary file 1A*) as a translational fusion to a 9 Ala linker and 2 × GFP followed by *OCS* terminator sequences. When transformed into *kan1-2 kan2-1* segregating plants, this construct complements the mutant phenotype. The triple marker line was generated by transforming *KAN1p:: KAN1−2 × GFP* into a *REVp::REV-2 × Ypet; PIN1p::PIN1-CFP* transgenic line. *KAN1p:: KAN1−2 × CFP* or *KAN1p::KAN1−2 × Ypet* containing a fusion to 2 copies of CFP or Ypet, respectively, were constructed similarly. *KAN1p::KAN1−2 × Ypet* and *PIN1p::PIN1-GFP* were combined in T-DNA vector *BGW* (*Karimi et al., 2002*) by Gateway technology (Invitrogen) for generation of a double marker transgenic line.

The *KAN1* cDNA was amplified by PCR with primers K1 cDNA F and K1 cDNA R to generate C-terminal translational fusion to a 9 Ala linker followed by single GFP or 2 × GFP followed by pea *rbcS E9* terminator sequence (*Zuo et al., 2001*) and cloned into the pOp6/LhGR two-component system (*Craft et al., 2005*) for dexamethasone-inducible misexpression.

An *ATML1p::LhGR* driver containing 3.4 kb of the L1-specific *ATML1* gene (At4g21750) fused to the chimeric LhGR transcription factor and a *6Op::KAN1-GFP* expression construct in a *pSULT* sulfadiazine-resistant T-DNA vector (*ATML1 >>KAN1* GFP) was generated. The *pSULT* T-DNA vector was derived from *pMLBART* by replacing the *NOSp::BAR* gene with *1'−2'p::*SUL (*Rosso et al., 2003*), a plant selectable marker that confers resistance to sulfadiazine herbicide to create *pSULT*. A *CLV3p:: LhGR* driver containing 1.49 kb of upstream regulatory sequences was PCR amplified with primers CLV3p F and CLV3p R along with 1.35 kb of downstream regulatory sequences with primers CLV3utr F and CLV3utr R was combined with *6Op::KAN1−2 × GFP* in *pSULT* T-DNA vector (*CLV3 >>KAN1-2×GFP*).

*ATML1 >>REVr-2×VENUS* is a sulfadiazine-resistant T-DNA vector to misexpress microRNA resistant REV-2 ×VENUS fusion, where *6Op::REVr-2×VENUS* was constructed by cloning a 1148 bp *Bam*HI-*Xcm*I microRNA resistant REV cDNA (a gift from J. Bowman) harbouring two previously characterized silent mutations that disrupt the binding of MIRNA 165/166 to the coding sequence of *REV* as previously described (*Emery et al., 2003*) downstream of the *6Op* and in frame with the wild type *REV-2 ×VENUS* coding sequences. *miR166Ap::GFPER* T-DNA construct was kindly provided by K. Nakajima (*Miyashima et al., 2011*). The MIR165/166 biosensor was created based on the design presented by Smith Z. R. et al. (*Smith and Long, 2010*) in the AlcR/AlcA expression system (*Roslan et al., 2001*) for ethanol-inducible expression. The sequences conferring MIR165/166 sensitivity from the *REV* coding sequence (*REV*) and the sequences conferring MIR165/166 insensitivity (REVr) were fused to *mCherry* (*Shaner et al., 2004*) with endoplasmic reticulum localization sequences *mCherryER*, which was synthesized de novo (Genscript). The MIR165/166-*mCherryER* biosensors (both biosensor and control) were cloned as *Hind*III-*Bam*HI fragments downstream of the *AlcA* regulatory sequences in the *UBQ10p:AlcR_BJ36* plasmid vector. *UBQ10p:AlcR* was constructed by cloning the *UBQ10* promoter 2 kb fragment upstream of *AlcR* and the *OCS* terminator. Both *UBQ10p:: AlcR* and *AlcA::REV-mCherryER* or *AlcA::REVr-mCherryER* components were combined in the T-DNA vector *pMOA34*.

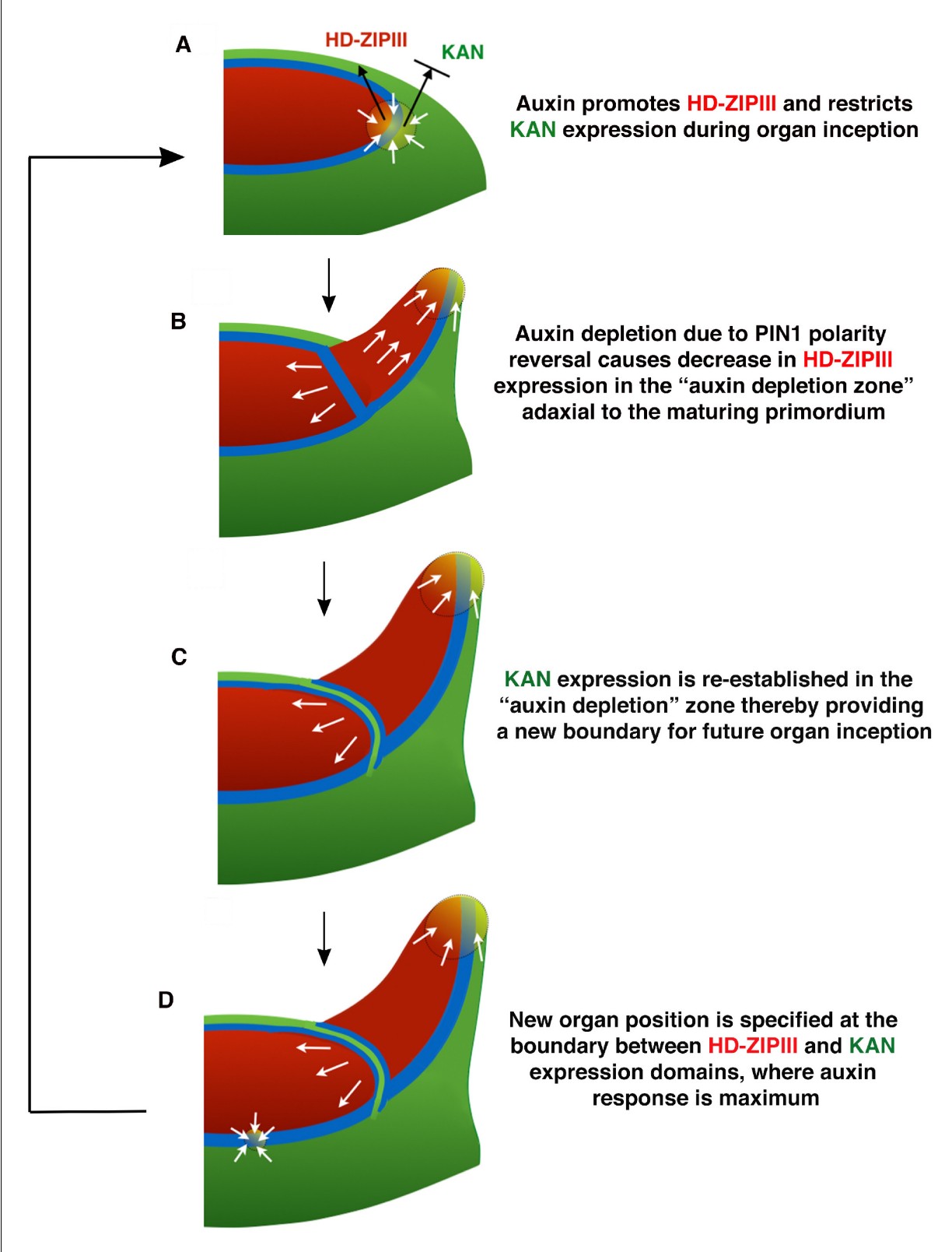

**Figure 14.** Conceptual Model. (**A**) During organ inception, PIN1 polarities (white arrows) and the alignment of microtubule arrays converge to create an auxin maximum and promote growth oriented towards the epidermal boundary between HD-ZIPIII and KANADI (KAN) expression domains. As auxin accumulates, it promotes the expression of HD-ZIPIII thereby resulting in its extension towards the PIN1 convergence site. At the same time, auxin also prevents the acropetal expansion of KAN. Thus the boundary becomes fixed to the underlying cells. (**B**) As the primordium grows, PIN1 polarity in cells

*Figure 14 continued on next page*

*Figure 14 continued*

adaxial to the primordium reverse towards the meristem center and adjacent incipient primordia, thereby creating an auxin depletion zone leading to a reduction in HD-ZIPIII expression. (**C**) The reduction in auxin results in the re-establishment of KAN expression between the meristem and organ. (**D**) Auxin in the vicinity of the boundary in adjacent tissues leads to a localized transcriptional response that orients the polarity of surrounding cells into a convergence pattern, most likely via mechanical signals (*Bhatia et al., 2016*).

DOI: https://doi.org/10.7554/eLife.27421.026

The *WOX1p::2 × GFP-WOX1* construct in *pMLBART* T-DNA vector was generated as follows: 2.2 kb of *WOX1 (At3g18010)* upstream promoter sequence was amplified with primers WOX1p F and WOX1p R and cloned using restriction enzymes KpnI and BamHI. 3.6 kb of *WOX1* coding sequence plus 1.65 kb 3'-regulatory sequences was amplified from wild-type Col-O genomic DNA with the primers WOX1g F and WOX1g R and cloned using restriction enzymes *Bgl*II and *Spe*I. 2 copies of GFP were inserted in frame at the start of the *WOX1* coding sequence at the *Bam*HI and *Bgl*II sites. A double marker was generated by transforming the *WOX1p::2 × GFP-WOX1* into a *PIN1p::PIN1-*CFP transgenic line.

The *PRSp::PRS-2×GFP* construct in *pMOA34* T-DNA vector was made by amplification of 3.9 kb *PRS (At2g28610)* genomic sequence (similar to [*Shimizu et al., 2009*]) with primers PRSg F and PRSg R to create a C-terminal fusion to *2 × GFP* followed by *OCS* 3'regulatory sequences. Marker combinations were generated by transforming the *PRSp::PRS-2×GFP* into either a *PIN1p::PIN1-CFP* transgenic line or into *REVp::REV-2×YPET PIN1p::PIN1-CFP* line. (*PRSp::PRS-2×GFP*) and (*KAN1p:: KAN1−2 × CFP*) were combined in T-DNA vector BGW (*Karimi et al., 2002*) by Gateway technology (Invitrogen) for generation of a double marker transgenic lines.

A short tandem target mimic (STTM) construct to target MIR165/166 (*Yan et al., 2012*) was generated in the *pOp6/LhGR* two-component system for dexamethasone-inducible expression with a *UBQ10p::GRLh* driver. *STTM MIR165/166-88* sequence (*Yan et al., 2012*) was synthesized de novo (Genscript) and cloned downstream of *6 × Op* to create *6 × Op::STTM 165/166*. Both components were combined in a sulfadiazine T-DNA *pSULT* (*UBQ10 >>STTM 165/166*).

*ATML1 >>PHVr* is a sulfadiazine-resistant T-DNA vector containing a mutated version of *PHV* cDNA (a gift from J. Bowman) with a Gly to Asp amino acid change that disrupts the miRNA165/166 binding site in the *PHV* gene (*McConnell et al., 2001*). *6 × Op::PHVr* was constructed by cloning a 2.6 kb *Xho*I-*Bam*HI *PHVr* cDNA downstream of *6 × Op* and upstream of *pea3A* terminator sequences. *ATML1 >>PHVr* was transformed into a *REVp::REV-2×YPet; PIN1p::PIN1-CFP* transgenic line.

*DR5−3 × VENUS v2* reporter gene (*Liao et al., 2015*) was a generous gift from Prof Dolf Weijers (Wageningen University). The line *R2D2 PIN1p::PIN1-GFP* was described previously (*Bhatia et al., 2016*). The seeds for *pin1-4* mutant were a kind gift from Prof. David Smyth (Monash University) and the heterozygote plants were crossed wild type plants carrying *pREV::REV-2xVENUS* and *pKAN:: KAN-2xGFP*. *pCLV3::GFP-ER* has been described previously (*Gordon et al., 2009*).

## Dexamethasone induction

For inducible gene perturbations in the vegetative SAM, seeds were germinated directly on GM medium containing 10 μM Dexamethasone (Sigma, stock solution was prepared in Ethanol). Seedlings were then dissected for imaging at 4 DAS, 5 DAS or 7 DAS depending on the experiment. For DEX induction in the IM, 10 μM DEX solution containing 0.015% Silwet L-77 was applied to the IM every second day three times. Inflorescences were then dissected and imaged. The number of T2 inducible transgenic lines that exhibit the presented phenotypes and the frequencies of phenotypes amongst imaged plants is shown in *Supplementary file 1B* and associated caption.

## Confocal microscopy

Plants were dissected for imaging as previously described (*Bhatia et al., 2016*; *Heisler and Ohno, 2014*) and imaged with a Leica SP5 Upright confocal microscope using an Argon laser. The objective used was a water-immersion HCX IRAPO L25x/0.95 W (Leica). Excitation for CFP is 458 nm, GFP is 488 nm, YFP (YPet and VENUS) is 514 nm and tdTomato is 561 nm. Emission signal were collected at 460–480 nm for CFP, 490–512 nm for GFP, 516–550 nm for YFP (YPet and VENUS), and 570–620 nm for tdTomato. The resulting z-stacks were processed using the Dye Separation (Channel mode or

automatic mode) function available in the LAS AF program in order to separate the GFP channel from the YFP (YPet or VENUS) channel. Three software packages: LAS AF from Leica, Imaris 8.0.2 by Bitplane and FIJI (https://fiji.sc) were used for data analysis. Ratios for R2D2 were calculated as described previously (*Bhatia et al., 2016*).

## Measurement of distance between organs
For distance measurements between oppositely positioned leaves on plants transgenic for inducible KAN1 (*CLV3 >>KAN1-2×GFP* – see above) the measurement tool from Imaris 8.0.2 (Bitplane) was used. For comparisons to control, untreated seedlings grown on GM were compared to seedlings grown on DEX for 5 days. t-Test was performed using Excel.

## Chemical treatments for auxin depletion in the inflorescence meristems
500 mM stock solutions of auxinole, yucasin and L-Kynurenine were prepared separately in DMSO. The stocks were diluted in 1 mL 0.1M phosphate buffer in sterile water to make a working solution containing all the three drugs (0.2 µL of each stock) to a final concentration of 100 mM each. The final concentration of DMSO in the working solution containing all the three drugs was 0.06%.

Treatments were carried out on the inflorescence meristems of whole plants transplanted from soil to boxes containing GM medium supplemented with vitamins. The older flowers were removed as described (*Heisler and Ohno, 2014*). The plants were chosen such that the stem of the meristem was a few millimeters above the rosette to prevent the drug solution from dispersing into the surrounding medium. After imaging, the meristems were carefully dried using a thin strip of sterile filter paper to remove excess water. Approximately 50 µL of the drug solution was added directly to the meristem, drop-wise. The meristems were treated only once in a time course of 12–18 hr.

## NPA treatment on seedlings carrying R2D2 and DR5 markers
Seedling aged 3 DAS were dissected to expose the meristem and the first leaf pair as described (*Bhatia et al., 2016*). After imaging, seedlings were transferred to new GM medium containing plates and blotted dry with thin strips of sterile paper. 5–10 µL of 100 µM NPA in sterile water (100 mM stock in DMSO) was added directly to the dissected seedlings every twenty-four hours for three days in total.

## Auxin treatment on seedlings carrying PRS and WOX1 markers
Seedlings aged 4DAS (days after stratification) were dissected to expose the meristem and the first leaf pair as described (*Bhatia et al., 2016*). 5 mM NAA (1M stock in 1M KOH) solution was prepared in liquid GM medium. Seedlings were then immersed in 100 µL of NAA containing medium in individual wells in 96 well plate and grown under continuous light without shaking for 12 hr.

## NAA, NPA and combined NAA and NPA treatments
For treatments on the vegetative meristems, 3DAS old seedlings expressing REV, KAN and PIN1 markers were dissected and imaged first. Samples were then treated with a combined solution of 100 µM NPA and 5 mM NAA (both diluted in water) or 100 µM NPA alone or 5 mM NAA alone. Samples were imaged and treated every 24 hr for 72 hr in total. For treatments on the infloresecnce meristems, flowering plants expressing REV, KAN1 and PIN1 markers and CLV3 GFP-ER marker were transferred from soil to boxes containing GM vit medium as described above. The infloresecnce meristems were dissected to remove older flower buds covering the meristem. The meristems were imaged, dried and treated with 10 µL of combined solution of 100 µM NPA and 5 mM NAA. Plants were imaged at the required timepoints (6 hr, 15 hr and 48 hr). Plants were treated with NAA and NPA after every timepoint imaging.

## Local application on *pin1-4* mutants
*pin1-4* mutant plants expressing REV and KAN1 markers with visibly smooth pin meristems were transferred from soil to boxes containing GM-vitamin medium. Leaves covering the meristems were dissected away. Control paste was applied locally to follow the meristem orientation over time. The meristems were imaged, dried and lanolin paste carrying 5 mM IAA (3-Indoleacetic acid) was administered locally at the meristem periphery. Plants were then imaged 15–17 hr later to follow REV

response at the site of auxin paste application. For this, first a z-stack representing an overview of the meristem was recorded. The paste was then scrapped off using a microfiber of the blotting paper carefully without damaging the meristem. A z stack was recorded again to follow REV and KAN expression at the site of paste application. A separate set of treated plants were imaged 24 hr later to follow REV and KAN expression in the grown out primordia at the site of auxin paste application.

## Pulsed laser ablations

Laser ablations on the inflorescence meristems were carried out using the Mai Tai multi-photon laser from Spectra Physics, which was controlled with LEICA SP5 confocal software. Z-stacks were acquired prior to ablation. Single cells were targeted one after the other using bleach point mode. Ablations were carried out at 800 nm with an output power of ~3W. Each pulse was shot for 1–15 milliseconds. Usually ablations were accomplished within 1–3 bursts of the laser. Ablated cells could be visually identified as their nuclei exploded resulting in unusual auto fluorescence. Z stacks were acquired immediately after the ablations.

## Auxin and auxin plus NPA combined treatments on ablated inflorescence meristems

After ablations, the meristems were carefully blotted dry using thin strips of sterile filter paper. 20 µL 5 mM NAA in sterile water (0.5M stock in 1M KOH) or 20 µL of a solution containing 5 mM NAA and 100 µM NPA in sterile water (100 mM NPA stock in DMSO) or mock solution were added directly to the meristems every 24 hr for 48 hr in total.

## Model for auxin and PIN1 dynamics

We developed a computational model to understand how interplay between the HD-ZIPIII and KAN pattern and PIN1 dynamics can influence auxin transport and primordia initiation. The model introduces a dependence on KANADI and REVOLUTA into previous models describing PIN1 and auxin dynamics (*Bhatia et al., 2016*; *Heisler et al., 2010*; *Heisler and Jönsson, 2006*; *Jönsson et al., 2006*; *Sahlin et al., 2009*). In the model, auxin resides in cell compartments and is able to move between cells either via active transport mediated by PIN1 proteins or passively via a diffusion-like process. PIN1 proteins cycle between cytosol and membrane compartments and a quasi-equilibrium model is used for determining its membrane localization at any time point. Auxin generates a signal able to polarize PIN1 in neighboring cells, i.e. a high auxin concentration increases the amount of PIN1 proteins in the neighboring cell membrane facing that cell. The molecule X in our model acts as a mediator of the signalling between auxin and PIN1, and the signal has previously been interpreted as a molecular (*Jönsson et al., 2006*), or mechanical stress signal (*Heisler et al., 2010*). In the model, the signal X is activated by auxin, and repressed by KAN and REV. The equations governing the dynamics of the molecules are

$$\frac{dA_i}{dt} = c_A - d_A A_i + \frac{1}{V_i}\left[ D \sum_{j \in \{N_i\}} a_{ij}(A_j - A_i) + T \sum_{j \in \{N_i\}} a_{ij}(P_{ji}A_j - P_{ij}A_i) \right],$$

$$\frac{dP_i^{[tot]}}{dt} = c_P - d_P P_i^{[tot]},$$

$$\frac{dX_i}{dt} = V_X \frac{A_i^{n_{XA}}}{\left(K_{XA}^{n_{XA}} + A_i^{n_{XA}}\right)} \frac{K_{XR}^{n_{XR}}}{\left(K_{XR}^{n_{XR}} + R_i^{n_{XR}}\right)} \frac{K_{XK}^{n_{XK}}}{\left(K_{XK}^{n_{XK}} + K_i^{n_{XK}}\right)} - d_X X_i,$$

where $A_i$ is the auxin concentration and $X_i$ is the level of the signalling molecule in cell $i$. $N_i$ is the set of cells neighboring cell $i$, $V_i$ is the cell volume of cell $i$, and $a_{ij} = a_{ji}$ is the cell wall area for the wall section between cells $i$ and $j$. $P_i^{[tot]}$ is the total PIN1concentration in the cytosol and membrane compartments of cell $i$. Membrane-bound PIN1 appears in the equation as $P_{ij}$, which is the PIN1 concentration in the membrane compartment of cell $i$ that faces cell $j$. A simple linear feedback between

the signal $X_j$ and $P_{ij}$ is used and a quasi-stable assumption, introduced in *Jönsson et al. (2006)*, leads to

$$P_{ij} = \frac{P_i^{[tot]}\left[(1-k_p)+k_pX_j\right]}{f_p + \sum_{k\in\{N_i\}}\left[(1-k_p)+k_pX_k\right]},$$

where $f_p = k_n/k_x$ is the ratio of endocytosis and exocytosis rates and $k_p$ sets the relation between symmetric and polarized exocytosis ($dP_{ij}/dt = k_x[(1-k_p)+k_pX_j]P_i - k_nP_{ij}$, where $P_i$ is the PIN1 in the cytosol compartment). The feedback between auxin and PIN1 is identical to previous models if the KAN and REV factors in the $dX_i/dt$ equation are 1 (e.g. by setting KAN and REV to zero in all cells), while the polarising signal becomes reduced in regions where KAN and/or REV are expressed. This effect tunes the interaction between the dorsoventral patterning and PIN1/auxin dynamics (cf. *Figure 8B and D*).

The model was simulated using the in-house developed software tissue (available at the gitlab repository https://gitlab.com/SLCU/TeamHJ/tissue; please see README.txt file in *Supplementary file 2* for details). A copy is archived at https://github.com/elifesciences-publications/TeamHJ-tissue. Files defining the model with parameter values (*Supplementary file 1C*) and initial configuration of (static) cell geometries and KAN and REV expression domains are provided in *Supplementary file 2*. The simulations use a 5th order Runge-Kutta solver with adaptive step size (*Press et al., 2007*), and initial auxin, PIN1 and X concentrations are set to zero in all compartments.

## Generation of geometrical template

The model defined above was run on a template containing a predefined KAN/REV pattern (provided as *Supplementary file 1C*, *Figure 9C*). The geometry of the template was generated by a combination of cell/wall growth and mechanical interactions together with a shortest path division rule (*Sahlin and Jönsson, 2010*). A KAN/REV pattern was generated by the equations

$$\frac{dK_i}{dt} = V_K\frac{r_i^{n_K}}{K_K^{n_K}+r_i^{n_K}} - d_KK_i,$$

$$\frac{dR_i}{dt} = V_R\frac{r_i^{n_R}}{K_R^{n_R}+r_i^{n_R}} - d_RR_i.$$

This system was run to equilibrium on the above-mentioned template. In the above equations $r_i$ is the distance of cell $i$ from the center of the template. The parameters were set to $V_K = V_R = d_K = d_R = 1$, $K_K = 30$, $K_R = 25$, $n_K = n_R = 20$. These parameters are set such that two distinct domains are created, with a small overlap of low KAN and REV concentrations in the boundary between these regions. To make the transition of KAN concentrations between domains sharper, KAN concentrations were set to 0 (1) if $K_i \leq 0.5$ ($>0.5$).

## Real time PCR

4 day-old wild-type Ler seedlings were immersed into 5 mM NAA solution in liquid GM medium, and grown under continuous light without shaking for 15 hr. Cotyledons, hypocotyl, and roots were removed under dissecting scope, and only shoot meristem and the first pair of leaves were collected and immediately frozen in liquid nitrogen. Each biological replicate, represents tissue from 10 to 15 individual seedlings. Five biological replicates were collected for both mock and NAA treatment. RNeasy Mini kit (Qiagen) was used according to manufacturer's instruction for RNA extraction. 1 microgram of RNA was used for cDNA preparation using Super script III reverse transcriptase for Q-PCR analysis. Q-PCRs were performed in a StepOne Plus Real Time PCR system thermo cycler (The applied bio systems) using 20 μl of PCR reaction containing 10 μL of SYBR Green mix (Roche), 1 μl of primer mix (10 μm), 2 μl of 1:10 diluted cDNA and 7 μl of water. Transcript levels were normalized to ACTIN2 transcript levels. Data was analyzed using the $2-\Delta\Delta CT$ method. A freely available online tool was used for analysis using an unpaired t-test of the RT-PCR results: http://graphpad.com/data-analysis-resource-center/. For p-value calculation, data entry format with mean, SD and N was used. Measurements and calculations for all replicates are provided in *Figure 8—source data 1*.

## Acknowledgements

We thank M E Byrne for helpful feedback and ideas on the manuscript. We thank J Bowman for helpful discussions as well as for plasmids containing REVr and PHVr cDNAs. We thank D Weijers and C Y Liao for a plasmid containing pDR5v2::ntdTomato; K Nakajima for the T-DNA construct of miR166Ap::GFPER, Dr Atsushi Miyawaki from RIKEN Brain Science Institute for the VENUS fluorescent protein which was obtained through an MTA. We thank Professor Ken-ichiro Hayashi, Okayama University of Science for providing Auxinole. The EMM laboratory is supported by funds from the Howard Hughes Medical Institute and the Gordon and Betty Moore Foundation (through grant GBMF3406). The research leading to these results received funding from the Australian Research Council (MGH) and European Research Council under the European Union's Seventh Framework Programme (FP/2007–2013)/ERC Grant Agreement n. 261081 (MGH), as well as the People Programme (Marie Curie Actions) under REA grant agreement n. 255089 (PS). The work was also supported by: the European Molecular Biology Laboratory (XY, MPC, CO, PS, NB, HR and MGH); the EMBL International PhD Programme (XY, NB and MPC); Gatsby Charitable Foundation (GAT3395/PR4) (HJ) and Swedish Research Council (VR2013-4632) (HJ). The authors declare no competing financial interests. Supplement contains additional data and movie. We declare there are no competing interests.

## Additional information

### Funding

| Funder | Grant reference number | Author |
| --- | --- | --- |
| H2020 European Research Council | 261081 | Marcus G Heisler |
| Marie Curie Actions | 255089 | Pia Sappl |
| Gordon and Betty Moore Foundation | GBMF3406 | Elliot M Meyerowitz |
| Gatsby Charitable Foundation | GAT3395/PR4 | Henrik Jönsson |
| Swedish Research Council | VR2013-4632 | Henrik Jönsson |

The funders had no role in study design, data collection and interpretation, or the decision to submit the work for publication.

### Author contributions

Monica Pia Caggiano, Xiulian Yu, Conceptualization, Data curation, Formal analysis, Investigation, Visualization, Methodology, Writing—original draft, Writing—review and editing; Neha Bhatia, Conceptualization, Data curation, Formal analysis, Visualization, Methodology, Writing—original draft, Writing—review and editing; Andre´ Larsson, Software, Formal analysis, Investigation, Visualization, Methodology, Writing—original draft; Hasthi Ram, Investigation, Visualization, Methodology, Writing—original draft, Writing—review and editing; Carolyn K Ohno, Resources, Investigation, Methodology, Writing—original draft, Project administration; Pia Sappl, Investigation, Methodology; Elliot M Meyerowitz, Supervision, Funding acquisition, Project administration, Writing—review and editing; Henrik Jönsson, Conceptualization, Software, Supervision, Funding acquisition, Investigation, Visualization, Methodology, Writing—original draft, Project administration, Writing—review and editing; Marcus G Heisler, Conceptualization, Supervision, Funding acquisition, Methodology, Writing—original draft, Project administration, Writing—review and editing

### Author ORCIDs

Neha Bhatia iD http://orcid.org/0000-0002-2165-5183
Elliot M Meyerowitz iD http://orcid.org/0000-0003-4798-5153
Henrik Jönsson iD http://orcid.org/0000-0003-2340-588X
Marcus G Heisler iD http://orcid.org/0000-0001-5644-8398

Decision letter and Author response
Decision letter https://doi.org/10.7554/eLife.27421.030
Author response https://doi.org/10.7554/eLife.27421.031

## Additional files

### Supplementary files

• Supplementary file 1. (A) List of primers used for generating transgenic constructs and quantitative real-time PCR. (B) Frequencies of phenotypes amongst different transgenic plant lines described in the study. For *ATML1 >>REVr-2×VENUS* (n = 20, number of imaged specimens) and *UBQ10 >>STTM 165/166* (n = 12) transgenic plants, we used T3 generation plants for imaging that exhibited the reported phenotypes at a frequencies ranging from 70% to 90%. For *ATML1 >>KAN1* GFP transgenic plants, we imaged a particular T2 line that exhibited meristem arrest after induction at a frequency of approximately 98% (n = 10). For *CLV3 >>K2G* transgenic lines, among 12 different T2 lines that showed leaf morphology changes and meristem arrest, we used a particular line that produced more than two leaves after meristem arrest at a frequency of 96.8% (n = 65). For *ATML1 >>PHVr* transgenic lines, we imaged a line that exhibited arrested organogenesis at a frequency of 33% (n = 12). An absence of phenotype was generally associated with low levels of induced transgene expression. (C) List of parameter values used in simulations. We have used the values from *Heisler and Jönsson (2006)*, which are based on experimental estimates where applicable
DOI: https://doi.org/10.7554/eLife.27421.027

• Supplementary file 2. Computational model files (separate file as one zipped archive).
DOI: https://doi.org/10.7554/eLife.27421.028

• Transparent reporting form
DOI: https://doi.org/10.7554/eLife.27421.029

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
