## [Decision Letter]

[Editors’ note: this article was originally rejected after discussions between the reviewers, but the authors were invited to resubmit after an appeal against the decision.]

Thank you for submitting your work entitled "Cell type boundaries organize plant development" for consideration by *eLife*. Your article has been reviewed by three peer reviewers, one of whom is a member of our Board of Reviewing Editors and the evaluation has been overseen a Senior Editor. The reviewers have opted to remain anonymous.

Our decision has been reached after consultation between the reviewers. Based on these discussions and the individual reviews below, we regret to inform you that your work will not be considered further for publication in *eLife*.

The reviewers and editors all agreed that this work focused on a long–standing issue of organ formation in plants and that the manuscript, using beautiful imaging and side–specific markers, presented convincing and high quality evidence for the presence of the boundary within the SAM and the influence of the location and form of this boundary on the position of leaf initiation, leaf polarity and the resulting leaf form. While this was appreciated, there was also a strong feeling from several reviewers that much of this, beautiful as it is, was largely confirmatory. In addition, there were concerns that some manipulations had technical issues (notably the long treatment with cocktails of auxin drugs) or use of multiple types of meristems that made the results somewhat ambiguous.

Reviewer #1:

Caggiano et al., address a long–standing issue of organ formation in plants. How do leaves acquire their characteristic arrangement around the shoot meristem and their flattened shape with different ad/abaxial domains? Is there a signal from the meristem, or are organs pre–patterned based on information in the meristem? Using detailed live–cell imaging and a number of genetic and experimental manipulations, the authors argue for pre–patterning in the meristem of ad/ab genes (primarily HD–ZIPs and Kanadi's) and for in instructive role of auxin in defining cell identity, potentially by influencing the expression domains of REV and KAN.

Overall, this work was interesting, the imaging of gene expression patterns was impressive and the overall view of pattern formation seems reasonable. I particularly like the extension of models from their previous SAM work to accommodate the new data provided here. Also, the phenotypes and diagrams explaining their potential origin in Figure 6 are quite cool! The authors point out alternative conclusions about D/V pattern that come from recently published work in tomato. I am afraid I don't have the background to evaluate the difference between these studies.

Reviewer #2:

Organ polarity is influenced by the organ placement relative to the shoot apical meristem (SAM). Several hypotheses have been proposed with respect to the nature of the positional information that directs organ polarity specification. One model suggests that pre–patterning in the SAM peripheral zone specifies the dorsal and ventral (abaxial and adaxial) sides of the leaf. In the current manuscript, using beautiful imaging and side–specific markers, the authors clearly show the existence of polarity within the peripheral zone of the SAM. They further show that the site of organ initiation is determined by the location of this dorsal–ventral boundary within the SAM. Alteration of this boundary by manipulation of the expression domain of KANADI, which promotes ventral identity, influences the initiation site. They further analyze the effect of auxin treatment and wounding on the placement of the boundary.

The manuscripts presents convincing and high quality evidence for the presence of the boundary within the SAM and the influence of the location and form of this boundary on the position of leaf initiation, leaf polarity and the resulting leaf form. The question is whether these findings are mainly confirmatory or bring new biological insights. The dependence of organ initiation and blade outgrowth on dorsal–ventral polarity has been shown before. It also has been shown that there is preexisting polarity within the peripheral zone of the SAM, but this manuscript shows it more clearly, including the pre–existing polar expression of ventral and dorsal determinants. The causal effect of auxin on this boundary is less convincing, as it is hard to tell how a combined NPA and auxin treatment or a treatment of a combination of 3 auxin biosynthesis and response inhibitors affects the different aspects of the process. The effect of these treatments is shown after a relatively long time, and these effects on the localization of the examined markers could result from an effect on the domain patterning within the SAM rather than an effect on REV expression.

Reviewer #3:

Caggiano et al., address the question, how the dorso–ventral (DV) pattern of leaves is established. Using fluorescent protein reporters they show that DV patterning of leaves correlates with central–peripheral patterning of the shoot apical meristem (SAM). The experimental data related to this paragraph are shown in 5 figures (Figure 1–Figure 5), of which many details are not addressed in the text. The novelty of this part is limited, because similar data have been published earlier (e.g. McConnell et al., 2001; reviewed by Barton et al., 2009).

Subsequently, the authors provide evidence that HD–ZIPIII and KAN genes repress leaf primordium initiation in the vegetative SAM. However, the question whether this is a direct influence on organ initiation or an indirect influence on SAM maintenance was not answered, because SAM organization in the arrested plants was not analyzed. Similar results were obtained by Kerstetter et al., 2001.

In the next paragraph, the consequences of ectopic KAN expression in the central domain of the SAM are studied. Most seedlings initiate several abnormal organs before their growth is arrested. The distinct organ phenotypes observed can be explained by the orientation of boundaries within organ founder cells.

Furthermore, the manuscript focuses on the interplay between auxin, HD–ZIPIII–expression and KAN–expression. Experiments using either NAA and NPA or a combination of auxinole, yuccasin and kyn led to the conclusion that high levels of auxin promote REV expression, whereas depletion of auxin promotes KAN expression. In this context, the treatment with a combination of three different drugs, which is necessary to see the described extension of KAN expression, may also have a different effect on gene expression than the one mediated through the observed depletion of auxin. The interplay of auxin, HD–ZIPIII expression and KAN–expression has also been addressed in previous studies (e.g. Izhaki and Bowman, 2007; Ilegems et al., 2010) with similar results.

In the last paragraph, the authors show that wounding in inflorescence meristems (IMs) leads to a depletion of auxin around the wound followed by an accumulation of KAN, which could be eliminated by an NAA + NPA treatment. Again, part of these findings were already known (Heisler et al., 2010; Landrein et al., 2015). In addition, IMs may react different from vegetative meristems.

Taken together, the manuscript presents a nice set of experiments supporting the view that DV organization of leaves is pre–patterned by the expression domains of REV and KAN in the SAM. However, previous studies revealed many similar results. Therefore, the study by Caggiano et al., presents only an incremental advance in our understanding.

Aiming for an explanation of the origin of DV patterning of leaves, it seems problematic to mix results from experiments using vegetative meristems and using inflorescence meristems, because a flower primordium is not a structure showing a DV pattern like a leaf primordium.

The wounding experiments offer an alternative explanation for the observations made by I. Sussex, but we think they do not rule out the existence of a signal from the main meristem to the developing leaf primordium.

[Editors’ note: what now follows is the decision letter after the authors submitted for further consideration.]

Thank you for resubmitting your work entitled "Cell type boundaries organize plant development" for further consideration at *eLife*. Your revised article has been favorably evaluated by Christian Hardtke (Senior editor) and three reviewers (mix of new and previous), one of whom is a member of our Board of Reviewing Editors.

There was, as in the first version, considerable appreciation of the imaging quality and evidence for a prepattern mechanism. There was also concern that the role of auxin in the process was not strongly supported by the provided data. Specifically, there was continued concern about the length of auxin treatments and the potential for KAN and REV responses to be secondary. Additionally, because the highest levels of auxin–signaling activity are found in the boundary region, but REV is precisely not expressed there, it was unclear what REV induction there by NPA+NAA treatment means. Moreover, how would this observation relate to a potential auxin–mediated maintenance of REV expression in the central region of the meristem, where there appears to be very little auxin–signaling activity?

After considerable deliberation, we have decided to consider a revised manuscript, IF it can address this critical concern about the proposed mechanism of auxin–mediated pre–patterning via REV activation and KAN repression. The misexpression of boundary genes (reviewer 1) could be one way to add more evidence, but alternative experiments would also be acceptable.

A revised version should also take care to tone down the discussion related to the "Sussex signal" because those experiments were done in floral meristems where the D/V polarity is limited. Ideally the issue would be addressed experimentally (e.g. using tomato apices), but the authors could also consider simply rewriting this section.

Reviewer #1:

In this revision, the authors have addressed my concerns about data presentation and have clarified many of the confusing passages of the previous version.

In this current version, I see that they have generated a very high resolution description of organ boundary gene expression and have a plausible model for auxin (though like most auxin models, there are enough feedbacks that it’s hard to find a way to experimentally disprove it). To my mind a manipulation in which they misexpress boundary genes in such a way that polarity becomes locally reoriented (something like making fly wing clones that create artificial boundaries) would be the real test of their models. It seems to me that reagents to do these experiments should be available, and I would be very supportive of this manuscript if such data could be added.

Reviewer #2:

In the revised manuscript, Caggiano et al., address some of the concerns raised by the reviewers and explain and clarify the new findings presented in this paper. As before, the imaging is impressive and convincing with respect to the pre–patterning of in the polarity in the SAM and the causative role of the boundary in organ initiation.

The authors better clarify the reason for the use of multiple drugs to assess the role of auxin in the patterning. However, I still think that the presented data on the effect of auxin does not support the interpretation on the role of auxin in specifying the spatial patterning in the SAM periphery. The time from application to effect is still long, and the effect on KAN and REV may be secondary to other effects. Combined with the relatively broad application, the results in my opinion do not support the proposed mechanism of auxin–mediated pre–patterning via REV activation and KAN repression.

Reviewer #4:

This manuscript addresses a long–standing question in plant biology, the acquisition of dorso–ventral (DV; or adaxial/abaxial) polarity in plant leaves. The authors use fluorescence live imaging combined with transgenic manipulation of gene–expression domains and pharmacological treatments to characterize the role of the boundary region between dorsal and ventral gene–expression domains in the meristem in leaf–polarity acquisition. They confirm and provide further detail on the pre–pattern in the shoot meristem involving centrally expressed HD–ZIPIII and peripherally expressed KANADI1 genes, and show that the boundary between the two domains, where neither gene is expressed strongly, is the site of auxin convergence driven by PIN1 and the highest auxin–signaling activity. Overexpressing either REV or KAN1 throughout the meristem epidermis abolishes organ initiation, consistent with their documented role in suppressing auxin activity. Together this suggests that the auxin maxima can only form in the boundary domain to trigger organ outgrowth there. Forcing KAN1 expression in the meristem centre and thus creating a new DV boundary there can lead to leaf radialization or even polarity reversal, indicating that the pre–pattern in the meristem can have an instructive function for leaf DV polarity. Having demonstrated that the pattern of REV and KAN1 expression determines the pattern of auxin responsiveness in the meristem, the authors then ask whether auxin in turn influences the expression pattern of these genes. Treating meristems with NAA and NPA causes an increase in the REV expression domain into the boundary region and can prevent the re–establishment of KAN1 expression in the periphery separating an outgrowing primordium from the meristem; conversely, inhibiting auxin signaling leads to an encroachment of KAN1 expression towards the meristem centre. Lastly, the authors show that the previously demonstrated auxin depletion around wound sites leads to an upregulation of KAN1 expression around the wound. This latter observation is offered as an explanation for the leaf radialization seen after separating a leaf primordium from the meristem in Sussex' classical experiments.

How DV polarity in leaves is determined is a question of fundamental importance and broad interest in plant development. As the ongoing interest in the quest for the proposed Sussex signal indicates, the work presented is clearly highly timely. The manuscript provides a mix of rather confirmatory and novel results. Important novel aspects are the demonstration that disturbing the DV expression pre–pattern in the meristem can lead to leaf–polarity defects; the finding that auxin in turn can modulate this DV expression pre–pattern; and the observation that peripheral/abaxial identity is induced surrounding wounds in the meristem.

I have the following comments:

1) The finding of a likely causal role of the pre–pattern on leaf polarity is an important advance. Also, the observation that auxin can modulate the DV expression pattern is interesting, yet its biological significance appears somewhat unclear. As the authors show in Figure 1 and in their model in Figure 9, the highest levels of auxin–signaling activity are found in the boundary region. However, it is precisely there that REV is not expressed. Thus, what does it mean that REV can be induced there by NPA+NAA treatment? And how does this observation relate to a potential auxin–mediated maintenance of REV expression in the central region of the meristem, where there appears to be very little auxin–signaling activity? Have the authors attempted to incorporate the auxin effect on REV expression into their model? Does this provide any information about the possible significance of this interaction?

2) The alternative explanation offered for the effects of wounds separating a primordium from the meristem is novel, interesting and potentially of broad significance. However, its impact appears limited, as the KAN1 upregulation around the entire wound is currently only shown clearly for Arabidopsis inflorescence meristems (while the seedling meristems undergo more dramatic re–organization, making their behavior more difficult to interpret), where the action of a Sussex signal is hard to demonstrate due to the limited DV polarity of floral meristems. This means that the alternative explanation is not backed up by functional experiments. Ideally, the authors would directly test in a different system, such as the tomato apex, whether NAA+NPA treatment after appropriate wounding could prevent organ abaxialization.

---

## [Author Response]

[Editors’ note: the author responses to the first round of peer review follow.]

Reviewer #2:Organ polarity is influenced by the organ placement relative to the shoot apical meristem (SAM). Several hypotheses have been proposed with respect to the nature of the positional information that directs organ polarity specification. One model suggests that pre–patterning in the SAM peripheral zone specifies the dorsal and ventral (abaxial and adaxial) sides of the leaf. In the current manuscript, using beautiful imaging and side–specific markers, the authors clearly show the existence of polarity within the peripheral zone of the SAM. They further show that the site of organ initiation is determined by the location of this dorsal–ventral boundary within the SAM. Alteration of this boundary by manipulation of the expression domain of KANADI, which promotes ventral identity, influences the initiation site. They further analyze the effect of auxin treatment and wounding on the placement of the boundary.The manuscripts presents convincing and high quality evidence for the presence of the boundary within the SAM and the influence of the location and form of this boundary on the position of leaf initiation, leaf polarity and the resulting leaf form.

We thank the reviewer for these positive remarks.

The question is whether these findings are mainly confirmatory or bring new biological insights.

Although we agree that, in some respects, the paper presents confirmation of one theory over others regarding how polarity is established and the role of a “Sussex signal”, we feel that this is definitely not the whole story and in any case, that it is quite a big deal.

Firstly, the debate regarding a pre–pattern or inductive signal has been going on for 70 years and recent papers have highlighted the fact that the issue is still not resolved. For instance, an extensive review published just recently on this exact topic by Cris Kuhlemeier and Marja Timmermans in Development pushes the Sussex signal idea. In this review they “postulate possible candidates for the ‘Sussex signal’ – the elusive meristem–derived factor that first ignited interest in this important developmental problem.” (Kuhlemeier and Timmermans, 2016).

Regarding other novel findings, our results on the role of auxin on REV and KAN expression are entirely new and provide great clarity for understanding exactly how dorsoventrality is established. We show that although there is a pre–pattern of dorsoventral gene activity in the SAM, this pattern is highly dynamic, being dependent on changing auxin levels and only becomes anchored to the cells when high auxin levels are present at organ inception. This finding reveals that auxin not only acts to trigger organ growth but also, that it plays a key role in establishing patterns of organ differentiation and the continuance of the dorsoventral boundary itself. We now provide extra data to back these conclusions (see below) and highlight it more clearly in the text and with our new Figure 14. Our unique high resolution data also enables us to propose the first explicit model based on known molecular players to explain how these boundaries actually work. We have added text to the intro and discussion as well as a diagram (Figure 14) to highlight this point.

These points are reiterated in more detail below.

The dependence of organ initiation and blade outgrowth on dorsal–ventral polarity has been shown before. It also has been shown that there is preexisting polarity within the peripheral zone of the SAM, but this manuscript shows it more clearly, including the pre–existing polar expression of ventral and dorsal determinants.

Regarding the previously published work on organ initiation / blade outgrowth and DV polarity, the very fact that the pre–pattern issue has not been settled confirms to us that the clues gained from these studies haven’t provided the clarity we need in order to know what’s really going on. For instance, one of the main figures from the recent Kuhlemeier and Timmermans review (Figure 6) shows the rough expression domains of HD–ZIPIII and KAN in the SAM but the rest is simply a model they propose (Kuhlemeier and Timmermans, 2016). There are several errors including the proposal that KAN1 is excluded from the primordium initially. Many other reviews indicate no pre–pattern at all (e.g. (Kalve et al., 2014; Liu et al., 2012; Nakata et al., 2012). Even when PIN1 and REV were examined at high resolution together in our previous paper (Heisler et al., 2005), we could only suggest that the DV boundary may be positioned by PIN1. Only now, by looking at KAN1 and testing the role of the boundary directly are we able to demonstrate the opposite relationship, i.e. the boundary influences where PIN1 localizes auxin.

In terms of functional data, Izhaki et al., (2007) shows through genetics that both KAN and HD–ZIPIII activities are required for repressing ectopic organogenesis (Izhaki and Bowman, 2007). However, the explanation given does not mention dorsoventral boundaries. Rather, the authors propose that the KAN genes control PIN1 expression and therefore auxin flow and that the HD–ZIPIIIs are seen as necessary for SAM formation. Ectopic organs therefore form because of aberrant auxin transport patterns and a missing SAM. This paper was published after the polar expression domains of KAN1 and REV in embryos were reported and yet no link was made between the positioning of organs and DV boundaries because the exact spatial relationship between these factors and PIN1 was not examined.

In contrast, our paper not only reports how all relevant factors relate spatially, but also, that the boundaries of the SAM are contiguous with that of the leaf and that, just as ectopic expression of REV and KAN blocks leaf blade growth, these factors also block leaf initiation. This has not been reported previously. We think this in itself is a major finding because now we can start to think of these seemingly distinct development processes as being not so different and controlled by one master patterning mechanism.

Besides concretely linking the activity of DV boundaries to organ formation, the high–resolution data we provide also forms the basis of a new hypothesis for how these boundaries actually work. By being able to see that PIN1 initiates organs specifically in a small gap between the REV and KAN domains and linking this with our other novel finding that HD–ZIPIII activity blocks organ initiation, we were able to present the first explicit model based on known molecular players to account for DV boundary activity. We feel this is a major step forward since we now have a testable hypothesis to help understand boundary function.

Additional major findings not reported previously include:

1) A new relationship between auxin and cell differentiation. In our revised version, we have further highlighted this discovery by adding additional data showing that the changes to REVKAN patterning in the inflorescence meristem caused by high auxin levels are even more dramatic in the vegetative meristem, where the reestablishment of KAN1 expression between the meristem and leaf can be completely blocked (Figure 11).

2) Understanding how meristems reorganize themselves in response to wounds is a major question in plant development. So far, we know that factors such as WUS re–pattern themselves in response to wounds and we have models for how this occurs. However more recently it’s been discovered that wounding alters cell polarity and as a consequence, auxin becomes depleted around wounds. What has been missing then is an understanding of the functional consequence of this depletion. Our paper now reveals that such changes in auxin play a key role in enabling the meristem to re–pattern DV gene expression. This work also enables us to provide a novel, straightforward and parsimonious explanation for the observations that originally prompted Sussex’s inductive signaling hypothesis.

In summary, we feel strongly that the imaging data we provide does more than simply add detail to what is already published, it also reveals key features that lead to major new insights and discoveries.

The causal effect of auxin on this boundary is less convincing, as it is hard to tell how a combined NPA and auxin treatment or a treatment of a combination of 3 auxin biosynthesis and response inhibitors affects the different aspects of the process. The effect of these treatments is shown after a relatively long time, and these effects on the localization of the examined markers could result from an effect on the domain patterning within the SAM rather than an effect on REV expression.

Firstly, we note that there was an error in the legend of Figure 10 that stated that the combined drug treatment on the inflorescence meristem was over 48hrs. Instead, the period was 18 hours as we now make clearer in the text.

Secondly, we want to clarify that we do not try to make the case that auxin acts directly on REV. Rather, our aim is to show that auxin directly or indirectly promotes HD–ZIPIII expression and prevents KAN expression in cells not already expressing KAN. To build on this proposal we have now added additional data (Figure 10—figure supplement 1) showing that in the inflorescence meristem the response of REV and KAN1 to 2,4–D is identical to combined NAA and NPA treatment but over a shorter period of 24hrs. We have also tracked cells showing that while auxin promotes REV expression it does this only in cells not already expressing KAN (Figure 10). This leads to a firmer conclusion as to how polarity is established in organs and how this contrasts with what happens in between organs where auxin levels are low (see Discussion section).

Regarding the influence of the three drugs, we have added additional data to our paper showing that auxinole treatment alone causes a very mild version of the phenotype we see when adding all three drugs simultaneously (Figure 12). Importantly, as auxinole competes for the auxin binding pocket in TIR1, we think it makes perfect sense to add the auxin synthesis inhibitors as well to obtain a more extreme loss of auxin activity. We also note that the phenotype we obtain is exactly what you would might expect, i.e. low auxin levels and a “pin” apex.

Reviewer #3:Caggiano et al., address the question, how the dorso–ventral (DV) pattern of leaves is established. Using fluorescent protein reporters they show that DV patterning of leaves correlates with central–peripheral patterning of the shoot apical meristem (SAM). The experimental data related to this paragraph are shown in 5 figures (Figure 1–Figure 5), of which many details are not addressed in the text. The novelty of this part is limited, because similar data have been published earlier (e.g. McConnell et al., 2001; reviewed by Barton et al., 2009).

Although other studies have shown that the HD–ZIPIIIs are expressed centrally and that KAN1 and miR165/166 reporters are expressed peripherally, our aim here was to examine the precise spatio–temporal relationship between the patterning of these genes and organ initiation. In other words, where and when does organ initiation occur with respect to these patterns? None of the studies mentioned address this.

We find that organ initiation is centered precisely between the REV and KAN domains, which is a novel finding and suggests a pre–pattern that may function to control one aspect of organ positioning.

A similar effort to address the question of whether there is a pre–pattern or not for leaf dorsoventrality is the study by Tameshige et al., 2013 which addressed whether FIL is initially polarized with respect to leaf primordia or not. They concluded that FIL is initially not polar leading to the proposal that leaf primordia are not polarized to begin with (Tameshige et al., 2013). Our findings confirm their result for FIL but also show that FIL is the odd one out in this respect. We mention this just as an example to help illustrate that our findings with respect to leaf pre–patterning are novel. Further evidence is provided in or response to reviewer 2 above.

Subsequently, the authors provide evidence that HD–ZIPIII and KAN genes repress leaf primordium initiation in the vegetative SAM. However, the question whether this is a direct influence on organ initiation or an indirect influence on SAM maintenance was not answered, because SAM organization in the arrested plants was not analyzed. Similar results were obtained by Kerstetter et al., 2001.

Although we did not examine the expression of SAM markers, the central zone expresses HD–ZIPIII genes normally during its development. It is difficult to explain then how peripheral REV in the epidermis might shut down meristem growth. In fact, inflorescence apices ectopically expressing REV continue to grow vertically without a change in size as evidenced by the elongated “pin–like” phenotype. This also indicates that the SAM remains intact and functioning.

As for ectopic KAN1, we note in the paper that meristem arrest does occur. However, since we see the formation of a “pin” meristem when we activate KAN1 post–embryonically, this implies that organ formation stops prior to arrest of the meristem since stem tissue must have been produced while organs were not. We disagree with the reviewer that our results are similar to Kerstetter et al., (2001). Kerstetter et al., 2001 found that ectopic KAN1 gave rise to seedlings completely lacking a SAM (Kerstetter et al., 2001). This contrasts with our “pin” phenotype. However, this is not surprising given the contrasting methods to induce KAN1, i.e. our study does it post–embryonically and only in the epidermis.

In the next paragraph, the consequences of ectopic KAN expression in the central domain of the SAM are studied. Most seedlings initiate several abnormal organs before their growth is arrested. The distinct organ phenotypes observed can be explained by the orientation of boundaries within organ founder cells.

We are unsure how to respond to this as this comment corresponds to our stated conclusion.

Furthermore, the manuscript focuses on the interplay between auxin, HD–ZIPIII–expression and KAN–expression. Experiments using either NAA and NPA or a combination of auxinole, yuccasin and kyn led to the conclusion that high levels of auxin promote REV expression, whereas depletion of auxin promotes KAN expression. In this context, the treatment with a combination of three different drugs, which is necessary to see the described extension of KAN expression, may also have a different effect on gene expression than the one mediated through the observed depletion of auxin.

Please see our response to reviewer 2 with regards to the auxin and drug treatments.

The interplay of auxin, HD–ZIPIII expression and KAN–expression has also been addressed in previous studies (e.g. Izhaki and Bowman, 2007; Ilegems et al., 2010) with similar results.

Neither of these papers examined the response of REV and KAN1 to auxin. Nor did they examine the response of auxin to ectopic REV. Rather, they examined the ectopic production of organs in HD–ZIPIII and KAN mutants (Izhaki and Bowman, 2007) as well as the response of vascular tissues to loss and gain of KAN expression (Ilegems et al., 2010). Regarding the Izhaki paper, please see our response to reviewer 2 above where we discuss this paper in detail. For the second paper, there is no overlap in experiments or major conclusions since this study focused on vascular formation.

In the last paragraph, the authors show that wounding in inflorescence meristems (IMs) leads to a depletion of auxin around the wound followed by an accumulation of KAN, which could be eliminated by an NAA + NPA treatment. Again, part of these findings were already known (Heisler et al., 2010; Landrein et al., 2015). In addition, IMs may react different from vegetative meristems.

The major point of our ablations is not to demonstrate auxin depletion. It is to demonstrate a link between wound–induced auxin depletion and changes in REV and KAN1 expression. We start by citing those exact papers (Heisler et al., 2010; Landrein et al., 2015) as support for our hypothesis. We then repeat the experiment to monitor auxin depletion simply as a control to make sure the wounds are having the effect we expect in our particular context (i.e. without prior NPA treatment). Our major findings on the influence of wounds on REV and KAN expression and the role of auxin in this are completely novel.

Taken together, the manuscript presents a nice set of experiments supporting the view that DV organization of leaves is pre–patterned by the expression domains of REV and KAN in the SAM. However, previous studies revealed many similar results. Therefore, the study by Caggiano et al., presents only an incremental advance in our understanding.

Please see our response to reviewer 2. The fact that the issue of a Sussex signal is still a major controversy after 70 years demonstrates that our study is highly relevant. We argue that our results are also conclusive in terms of the pre–pattern and further provide an explanation for the original observed wounding responses. Our paper also contains many other major new findings as described in more detail above and now better highlighted in the new version of our manuscript.

Aiming for an explanation of the origin of DV patterning of leaves, it seems problematic to mix results from experiments using vegetative meristems and using inflorescence meristems, because a flower primordium is not a structure showing a DV pattern like a leaf primordium.

We apologize for the confusion. We should have used the term “floral bract” rather than primordium and we have now done so. All our results so far indicate that the establishment of polarity for the cryptic bract is almost identical to that of leaves consistent with published data indicating that the initial structure is indeed leaf–like rather than a floral meristem. The floral meristem develops some time later, after polarity for the bract has already been established and therefore we do not think it relevant. In any case, most of our results are presented for both the inflorescence and vegetative meristem contexts.

The wounding experiments offer an alternative explanation for the observations made by I. Sussex, but we think they do not rule out the existence of a signal from the main meristem to the developing leaf primordium.

At least for wounds, since KAN1 expression surrounds the wound regardless of its orientation with respect to the meristem, we maintain this is strong evidence against a meristem–oriented signal, as originally proposed by Sussex. The added evidence demonstrating the involvement of auxin provides a different and more parsimonious interpretation of Sussex’s experimental results. However, we agree that there are still important questions to be answered and write in our discussion that further experiments are required in tomato and other species.

[Editors’ note: the author responses to the re–review follow.]

Reviewer #1:In this revision, the authors have addressed my concerns about data presentation and have clarified many of the confusing passages of the previous version.In this current version, I see that they have generated a very high resolution description of organ boundary gene expression and have a plausible model for auxin (though like most auxin models, there are enough feedbacks that it’s hard to find a way to experimentally disprove it). To my mind a manipulation in which they misexpress boundary genes in such a way that polarity becomes locally reoriented (something like making fly wing clones that create artificial boundaries) would be the real test of their models. It seems to me that reagents to do these experiments should be available, and I would be very supportive of this manuscript if such data could be added.

We thank reviewer 1 for their positive comments. With regards to testing the model we propose using mosaics, we feel this is a major technical undertaking and would deserve to be presented in its own separate study. It is something we are working on for the future.

Reviewer #2:In the revised manuscript, Caggiano et al., address some of the concerns raised by the reviewers and explain and clarify the new findings presented in this paper. As before, the imaging is impressive and convincing with respect to the pre–patterning of in the polarity in the SAM and the causative role of the boundary in organ initiation.The authors better clarify the reason for the use of multiple drugs to assess the role of auxin in the patterning. However, I still think that the presented data on the effect of auxin does not support the interpretation on the role of auxin in specifying the spatial patterning in the SAM periphery. The time from application to effect is still long, and the effect on KAN and REV may be secondary to other effects. Combined with the relatively broad application, the results in my opinion do not support the proposed mechanism of auxin–mediated pre–patterning via REV activation and KAN repression.

We have now directly addressed these concerns with additional experiments. Firstly, we found that when auxin was applied to meristems on intact plants rather than plants in which the apices had been detached, the response of REV to exogenous auxin was much faster, starting within 6hrs (Figure 10—figure supplement 1) but becoming more obvious in less than 18hrs (Figure 10). Importantly, this exactly matches the wild type timing of events during regular development since REV expression visibly extends from the central zone into developing primordia a plastochron later than when PIN1 and DR5 mark these primordia (Heisler et al., 2005), i.e. around a 12hr delay.

As with the previous version of Figure 10, in the new Figure 10 we also show that auxin treatment inhibits the reestablishment of KAN1 in between primordia and the meristem (yellow and white arrows in Figure 10 compared to control K and L).

Regarding the timing for KAN1, during regular development, KAN1 expression is well established by stage P2 (Figure 1—figure supplement 1), around 1 plastochron (approx. 12hrs) after auxin is depleted in boundary regions at stage P1 (the primordia are staged according to Heisler et al., 2005 based on the time of PIN1 polarity reversal). Hence it is to be expected that any induction of KAN1 in response to low auxin levels will take a similar length of time, which is entirely consistent with the results of our auxin depletion experiments (i.e. 18hrs shown in Figure 12).

Our data also suggest that once auxin depletion has occurred, cells are committed to expressing KAN1 even if auxin levels subsequently increase. This is indicated by the finding that exogenous auxin treatment cannot prevent the onset of KAN1 expression if auxin is applied at the P1 stage (when auxin has already been depleted in the surrounding cells – blue arrowheads in Figure 10). This is in contrast to applying auxin at the i1 and i2 stages when PIN1 reversal and auxin depletion have not occurred yet (white and yellow arrowheads in Figure 10 compared to controls K and L).

To further show that auxin acts locally to influence the expression domains of both REV and KAN1 in the periphery we applied auxin locally to pin1 mutant meristems and found that the response of REV and KAN1 is also local (Figure 10—figure supplement 2).

Lastly, we now also present data showing that neither auxin addition nor depletion obviously influence meristem patterning overall as evidenced by the central zone marker CLV3, indicating a more direct influence on primordium patterning (Figure 10—figure supplement 3 and Figure 12—figure supplement 3).

Again, we want to emphasize that the conclusions of our paper do not depend on REV and KAN being direct auxin targets and therefore that they must respond extremely rapidly (e.g. 30mins) to auxin. Accordingly, given the timing, we have now explicitly stated in our discussion that the regulation of these genes by auxin is likely to be indirect (Discussion section). At the same time however, our data do show that auxin modulates their expression patterns locally in the periphery and reasonably rapidly, and that this regulation is relevant to wound response.

Reviewer #4:This manuscript addresses a long–standing question in plant biology, the acquisition of dorso–ventral (DV; or adaxial/abaxial) polarity in plant leaves. The authors use fluorescence live imaging combined with transgenic manipulation of gene–expression domains and pharmacological treatments to characterize the role of the boundary region between dorsal and ventral gene–expression domains in the meristem in leaf–polarity acquisition. They confirm and provide further detail on the pre–pattern in the shoot meristem involving centrally expressed HD–ZIPIII and peripherally expressed KANADI1 genes, and show that the boundary between the two domains, where neither gene is expressed strongly, is the site of auxin convergence driven by PIN1 and the highest auxin–signaling activity. Overexpressing either REV or KAN1 throughout the meristem epidermis abolishes organ initiation, consistent with their documented role in suppressing auxin activity. Together this suggests that the auxin maxima can only form in the boundary domain to trigger organ outgrowth there. Forcing KAN1 expression in the meristem centre and thus creating a new DV boundary there can lead to leaf radialization or even polarity reversal, indicating that the pre–pattern in the meristem can have an instructive function for leaf DV polarity. Having demonstrated that the pattern of REV and KAN1 expression determines the pattern of auxin responsiveness in the meristem, the authors then ask whether auxin in turn influences the expression pattern of these genes. Treating meristems with NAA and NPA causes an increase in the REV expression domain into the boundary region and can prevent the re–establishment of KAN1 expression in the periphery separating an outgrowing primordium from the meristem; conversely, inhibiting auxin signaling leads to an encroachment of KAN1 expression towards the meristem centre. Lastly, the authors show that the previously demonstrated auxin depletion around wound sites leads to an upregulation of KAN1 expression around the wound. This latter observation is offered as an explanation for the leaf radialization seen after separating a leaf primordium from the meristem in Sussex' classical experiments.How DV polarity in leaves is determined is a question of fundamental importance and broad interest in plant development. As the ongoing interest in the quest for the proposed Sussex signal indicates, the work presented is clearly highly timely. The manuscript provides a mix of rather confirmatory and novel results. Important novel aspects are the demonstration that disturbing the DV expression pre–pattern in the meristem can lead to leaf–polarity defects; the finding that auxin in turn can modulate this DV expression pre–pattern; and the observation that peripheral/abaxial identity is induced surrounding wounds in the meristem.I have the following comments:1) The finding of a likely causal role of the pre–pattern on leaf polarity is an important advance. Also, the observation that auxin can modulate the DV expression pattern is interesting, yet its biological significance appears somewhat unclear. As the authors show in Figure 1 and in their model in Figure 9, the highest levels of auxin–signaling activity are found in the boundary region. However, it is precisely there that REV is not expressed. Thus, what does it mean that REV can be induced there by NPA+NAA treatment? And how does this observation relate to a potential auxin–mediated maintenance of REV expression in the central region of the meristem, where there appears to be very little auxin–signaling activity? Have the authors attempted to incorporate the auxin effect on REV expression into their model? Does this provide any information about the possible significance of this interaction?

We thank reviewer 4 for their positive comments.

Regarding the location of auxin signalling and REV expression, we note that during normal development REV expression extends into the boundary region that normally separates REV and KAN1 a plastochron after PIN1 upregulation. We apologize as this was not properly described in association with Figure 1 previously and so we have now added more text and a sup figure to fully detail this observation (Figure 1—figure supplement. 1). As can be seen from the new supplementary figure, the size of the gap shrinks initially as REV expression increases and extends but then becomes more apparent at later stages. Hence our treatments with NPA+NAA recapitulate the initial extension and increase in REV expression observed during the early stages of primordium development.

What does this regulation by auxin mean? The first point of significance is that we show that auxin acts to establish as well as maintain proper DV cell type patterning in new primordia. This is a new role for auxin that is distinct to its well–known function in organ initiation. This role is difficult to uncover because if you take auxin away you don’t get a primordium. Only by depleting auxin after organs have initiated and observing with time–lapse are we able to come to this conclusion. The second point of significance is in relation to the next section where we link the regulation of REV/KAN by auxin to wounding.

Regarding the central zone, an important point is that while REV/KAN expression depends on auxin in the meristem periphery, our experiments show that in the central zone, this is not the case. REV is maintained in the central zone no matter how much we deplete auxin. Thus, we conclude that the regulation of REV/KAN in the meristem central zone occurs via a distinct, auxin independent pathway. Hence only once cells move out of the central zone into the periphery, do high auxin levels become important for maintaining REV and repressing KAN. We have tried to make this conclusion clearer in the relevant Results section and Discussion section.

Regarding incorporating auxin regulation of REV/KAN in the modelling, this has begun. However, the modelling is still inconclusive at this point since it lacks certain features we feel are necessary to fully test our ideas, such as localized cell proliferation.

2) The alternative explanation offered for the effects of wounds separating a primordium from the meristem is novel, interesting and potentially of broad significance. However, its impact appears limited, as the KAN1 upregulation around the entire wound is currently only shown clearly for Arabidopsis inflorescence meristems (while the seedling meristems undergo more dramatic re–organization, making their behavior more difficult to interpret), where the action of a Sussex signal is hard to demonstrate due to the limited DV polarity of floral meristems. This means that the alternative explanation is not backed up by functional experiments. Ideally, the authors would directly test in a different system, such as the tomato apex, whether NAA+NPA treatment after appropriate wounding could prevent organ abaxialization.

While we agree that further experiments in other species would increase the impact of the current work, we also feel that this would represent a major long–term project in itself. Therefore, at this stage, we have rewritten our discussion of these results to further highlight these concerns (Discussion section).